# Physics-Informed Regularization for Domain-Agnostic Dynamical System Modeling

**Zijie Huang**[1][*]  **Wanjia Zhao**[2][*][†]  **Jingdong Gao**[1]  **Ziniu Hu**[3]  **Xiao Luo**[1]
**Yadi Cao**[1]  **Yuanzhou Chen**[1]  **Yizhou Sun**[1]  **Wei Wang**[1]
[1]University of California Los Angeles, [2]Stanford University
[3]California Institute of Technology
https://treat-ode.github.io/

## Abstract

Learning complex physical dynamics purely from data is challenging due to the intrinsic properties of systems to be satisfied. Incorporating physics-informed priors, such as in Hamiltonian Neural Networks (HNNs), achieves high-precision modeling for energy-conservative systems. However, real-world systems often deviate from strict energy conservation and follow different physical priors. To address this, we present a framework that achieves high-precision modeling for a wide range of dynamical systems from the numerical aspect, by enforcing *Time-Reversal Symmetry* (TRS) via a novel regularization term. It helps preserve energies for conservative systems while serving as a strong inductive bias for non-conservative, reversible systems. While TRS is a domain-specific physical prior, we present the *first* theoretical proof that TRS loss can universally improve modeling accuracy by minimizing higher-order Taylor terms in ODE integration, which is numerically beneficial to various systems regardless of their properties, even for irreversible systems. By integrating the TRS loss within neural ordinary differential equation models, the proposed model TREAT demonstrates superior performance on diverse physical systems. It achieves a significant 11.5% MSE improvement in a challenging chaotic triple-pendulum scenario, underscoring TREAT's broad applicability and effectiveness. Code and further details are available at here.

## 1 Introduction

Dynamical systems, spanning applications from physical simulations (Kipf et al., 2018; Wang et al., 2020; Lu et al., 2022; Huang et al., 2023; Luo et al., 2023a; Xu et al., 2024; Luo et al., 2024) to robotic control (Li et al., 2022; Ni and Qureshi, 2022), are challenging to model due to intricate dynamic patterns and potential interactions under multi-agent settings. Traditional numerical simulators require extensive domain knowledge for design, which is sometimes unknown (Sanchez-Gonzalez et al., 2020), and can consume significant computational resources (Wang et al., 2024). Therefore, directly learning dynamics from the observational data becomes an attractive alternative.

Existing deep learning approaches (Sanchez-Gonzalez et al., 2020; Pfaff et al., 2021; Han et al., 2022a) usually learn a fixed-step transition function to predict system dynamics from timestamp $t$ to timestamp $t + 1$ and rollout trajectories recursively. The transition function can have different inductive biases, such as Graph Neural Networks (GNNs) (Pfaff et al., 2020; Martinkus et al., 2021; Lam et al., 2023; Cao et al., 2023) for capturing pair-wise interactions among agents through message passing. Most recently, neural ordinary differential equations (Neural ODEs) (Chen et al.,

---

[*]Equal contribution, Corresponding to Zijie Huang <zijiehuang@cs.ucla.edu>, Wanjia Zhao <wanjiazh@cs.stanford.edu>

[†]Work done as a visiting student at UCLA

38th Conference on Neural Information Processing Systems (NeurIPS 2024).

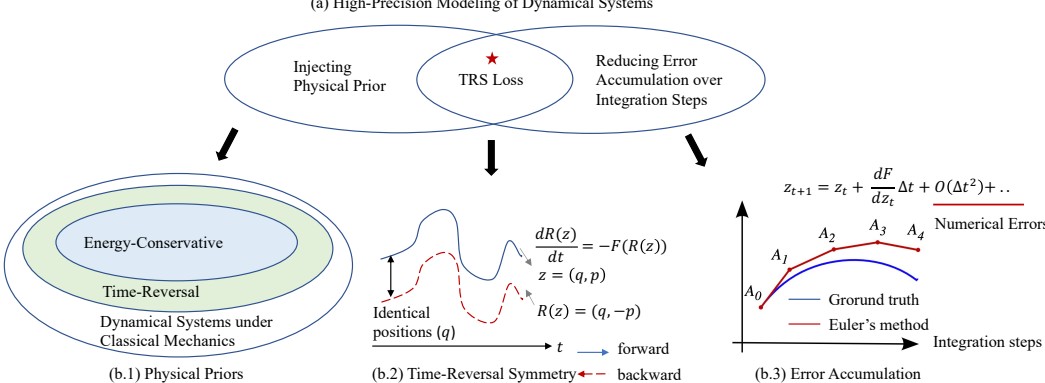

Figure 1: (a) High-precision modeling for dynamical systems; (b.1) Classification of classical mechanical systems based on (Tolman, 1938; Lamb and Roberts, 1998);(b.2) Tim-Reversal Symmetry illustration;(b.3) Error accumulation in numerical solvers.

2018; Rubanova et al., 2019) have emerged as a potent solution for modeling system dynamics in a continuous manner, which offer superior prediction accuracy over discrete models in the long-range, and can handle systems with partial observations. In particular, GraphODEs (Huang et al., 2020; Luo et al., 2023b; Zang and Wang, 2020; Jiang et al., 2023; Luo et al., 2023c) extend NeuralODEs to model interacting (multi-agent) dynamical systems, where agents co-evolve and form trajectories jointly.

However, the complexity of dynamical systems necessitates large amounts of data. Models trained on limited data risk violating fundamental physical principles such as energy conservation. A promising strategy to improve modeling accuracy involves incorporating physical inductive biases (Raissi et al., 2019; Cranmer et al., 2020). Existing models like Hamiltonian Neural Networks (HNNs) (Greydanus et al., 2019; Sanchez-Gonzalez et al., 2019) strictly enforce energy conservation, yielding more accurate predictions for energy-conservative systems. However, not all real-world systems strictly adhere to energy conservation, and they may adhere to various physical priors. Other methods that model both energy-conserving and dissipative systems, as well as reversible systems, offer more flexibility (Zhong et al., 2020; Gruber et al., 2024). Nevertheless, they often rely on prior knowledge of the system and are also limited to systems with corresponding physical priors. Such system diversity largely limits the usage of existing models which are designed for specific physical prior.

To address this, we present a framework that achieves high-precision modeling for a wide range of dynamical systems from the numerical aspect, by enforcing Time-Reversal Symmetry (TRS) via a novel regularization term. Specifically, TRS posits that a system's dynamics should remain invariant when time is reversed (Lamb and Roberts, 1998). To incorporate TRS, we propose a simple-yet-effective self-supervised regularization term that acts as a soft constraint. This term aligns *forward and backward trajectories* predicted by a neural network and we use GraphODE as the backbone. We theoretically prove that the TRS loss effectively minimizes higher-order Taylor expansion terms during ODE integration, offering a general numerical advantage for improving modeling accuracy across a wide array of systems, regardless of their physical properties. It forces the model to capture fine-grained physical properties such as jerk (the derivatives of accelerations) and provides more regularization for long-term prediction. We also justify our TRS design choice, showing case its superior performance both analytically and empirically. We name the model as TREAT (**T**ime-**Re**vers**a**l Symme**t**ry ODE).

Note that TRS itself is a physical prior, that is broader than energy conservation as depicted in Figure 1(b.1). It covers classical energy-conservative systems such as Newtonian mechanics, and also non-conservative, reversible systems like Stokes flow (Pozrikidis, 2001), commonly encountered in microfluidics (Kim and Karrila, 2013; Cao and Li, 2018; Cao et al., 2019). Therefore, TRS loss achieves high-precision modeling from both the physical aspect, and the numerical aspect as shown in Figure 1(a), making it domain-agnostic and widely applicable to various dynamical systems. We systematically conduct experiments across 9 diverse datasets spanning across 1.) single-agent, multi-agent systems; 2.) simulated and real-world systems; and 3.) systems with different physical

priors. TREAT consistently outperforms state-of-the-art baselines, affirming its effectiveness and versatility across various dynamic scenarios.

Our primary contributions can be summarized as follows:

- We introduce TREAT, a powerful framework that achieves high-precision modeling for a wide range of systems from the numerical aspect, by enforcing Time-Reversal Symmetry (TRS) via a regularization term.
- We establish the *first* theoretical proof that the time-reversal symmetry loss could in general help learn more fine-grained and long-context system dynamics from the numerical aspect, regardless of systems' physical properties (even irreversible systems). This bridges the specific physical implication and the general numerical benefits of the physical prior -TRS.
- We present empirical evidence of TREAT's state-of-the-art performance in a variety of systems over 9 datasets, including real-world & simulated systems, etc. It yields a significant MSE improvement of 11.5% on the challenging chaotic triple-pendulum system.

## 2 Preliminaries and Related Work

We represent a dynamical system as a graph $\mathcal{G} = (\mathcal{V}, \mathcal{E})$, where $\mathcal{V}$ denotes the node set of $N$ agents[3] and $\mathcal{E}$ denotes the set of edges representing their physical interactions. For simplicity, we assumed $\mathcal{G}$ to be static over time. Single-agent dynamical system is a special case where the graph only has one node. In the following, we use the multi-agent setting by default to illustrate our model. We denote $\boldsymbol{X}(t) \in \mathbb{R}^{N \times d}$ as the feature matrix at timestamp $t$ for all agents, with $d$ as the feature dimension. Model input consists of trajectories of feature matrices over $M$ historical timestamps $X(t_{-M:-1}) = \{\boldsymbol{X}(t_{-M}), \ldots, \boldsymbol{X}(t_{-1})\}$ and $\mathcal{G}$. The timestamps $t_{-1}, \cdots, t_{-M} < 0$ can have non-uniform intervals and take any continuous values. Our goal is to learn a neural simulator $f_\theta(\cdot): [X(t_{-M:-1}), \mathcal{G}] \to Y(t_{0:K})$, which predicts node dynamics $\boldsymbol{Y}(t)$ in the future on timestamps $0 = t_0 < \cdots < t_K = T$ sampled within $[0, T]$. We use $\boldsymbol{y}_i(t)$ to denote the targeted dynamic vector of agent $i$ at time $t$. In some cases when we are only predicting system feature trajectories, $\boldsymbol{Y}(\cdot) \equiv \boldsymbol{X}(\cdot)$.

### 2.1 NeuralODE for Dynamical Systems

NeuralODEs (Chen et al., 2018; Rubanova et al., 2019) are a family of continuous models that define the evolution of dynamical systems by ordinary differential equations (ODEs). The state evolution can be described as: $\dot{\boldsymbol{z}}_i(t) := \frac{d\boldsymbol{z}_i(t)}{dt} = g(\boldsymbol{z}_1(t), \boldsymbol{z}_2(t) \cdots \boldsymbol{z}_N(t))$, where $\boldsymbol{z}_i(t) \in \mathbb{R}^d$ denotes the latent state variable for agent $i$ at timestamp $t$. The ODE function $g$ is parameterized by a neural network such as Multi-Layer Perception (MLP), which is automatically learned from data. GraphODEs (Poli et al., 2019; Huang et al., 2020; Luo et al., 2023b; Wen et al., 2022; Huang et al., 2024) are special cases of NeuralODEs, where $g$ is a Graph Neural Network (GNN) to capture the continuous interaction among agents.

GraphODEs have been shown to achieve superior performance, especially in long-range predictions and can handle data irregularity issues. They usually follow the encoder-processor-decoder architecture, where an encoder first computes the latent initial states $\boldsymbol{z}_1(t_0), \cdots \boldsymbol{z}_N(t_0)$ for all agents simultaneously based on their historical observations as in Eqn 1.

$$\boldsymbol{z}_1(t_0), \boldsymbol{z}_2(t_0), ..., \boldsymbol{z}_N(t_0) = f_{\text{ENC}}(X(t_{-M:-1}), \mathcal{G}) \tag{1}$$

Then the GNN-based ODE predicts the latent trajectories starting from the learned initial states. The latent state $\boldsymbol{z}_i(t)$ can be computed at any desired time using a numerical solver such as Runge-Kuttais (Schober et al., 2019) as:

$$\boldsymbol{z}_i(t) = \text{ODE-Solver}(g, [\boldsymbol{z}_1(t_0), ...\boldsymbol{z}_N(t_0)], t) = \boldsymbol{z}_i(t_0) + \int_{t_0}^t g(\boldsymbol{z}_1(t), \boldsymbol{z}_2(t) \cdots \boldsymbol{z}_N(t)) dt. \tag{2}$$

Finally, a decoder extracts the predicted dynamics $\hat{\boldsymbol{y}}_i(t)$ based on the latent states $\boldsymbol{z}_i(t)$ for any timestamp $t$:

$$\hat{\boldsymbol{y}}_i(t) = f_{\text{DEC}}(\boldsymbol{z}_i(t)). \tag{3}$$

---

[3]Following (Kipf et al., 2018), we use "agents" to denote "objects" in dynamical systems, which is different from "intelligent agent" in AI.

However, vanilla GraphODEs can violate physical properties of a system, resulting in unrealistic predictions. We therefore propose to inject physics-informed regularization term to make more accurate predictions.

## 2.2 Time-Reversal Symmetry (TRS)

Consider a dynamical system described in the form of $\frac{d\boldsymbol{x}(t)}{dt} = F(\boldsymbol{x}(t))$, where $\boldsymbol{x}(t) \in \Omega$ is the observed states such as positions. The system is said to follow the *Time-Reversal Symmetry* if there exists a reversing operator $R : \Omega \mapsto \Omega$ such that (Lamb and Roberts, 1998):

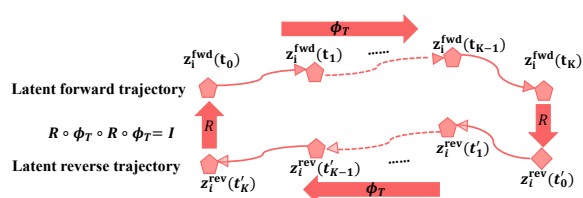

$$\frac{d\big(R \circ \boldsymbol{x}(t)\big)}{dt} = -F\big(R \circ \boldsymbol{x}(t)\big), \quad (4)$$

where $\circ$ denote the action of functional $R$ on the function $\boldsymbol{x}$.

Figure 2: Illustration of time-reversal symmetry based on Lemma 2.1. The total length of the trajectory is $t_K - t_0 = T$. $t'_k$ is the time index in the reverse trajectory, which points to the same time as $t_{K-k}$ in the forward trajectory.

Intuitively, we can assume $\boldsymbol{x}(t)$ is the position of a flying ball and the conventional reversing operator is defined as $R : \boldsymbol{x} \mapsto R \circ \boldsymbol{x}, R \circ \boldsymbol{x}(t) = \boldsymbol{x}(-t)$. This implies when $\boldsymbol{x}(t)$ is a forward trajectory position with initial position $\boldsymbol{x}(0)$, $\boldsymbol{x}(-t)$ is then a position in the time-reversal trajectory, where $\boldsymbol{x}(-t)$ is calculated using the same function $F$, but with the integration time reversed, i.e. $dt \mapsto d(-t)$. Eqn 4 shows how to create the reverse trajectory of a flying ball: at each position, the velocity (i.e., the derivative of position with respect to time) should be the opposite. In neural networks, we usually model trajectories in the latent space via $\boldsymbol{z}$ (Sanchez-Gonzalez et al., 2020), which can be decoded back to real observation state i.e. positions. Therefore, we apply the reversal operator for $\boldsymbol{z}$.

Now we introduce a time evolution operator $\phi_\tau$ such that $\phi_\tau \circ \boldsymbol{z}(t) = \boldsymbol{z}(t + \tau)$ for arbitrary $t, \tau \in \mathbb{R}$. It satisfies $\phi_{\tau_1} \circ \phi_{\tau_2} = \phi_{\tau_1 + \tau_2}$, where $\circ$ denotes composition. The time evolution operator helps us to move forward (when $\tau > 0$) or backward (when $\tau < 0$) through time, thus forming a trajectory. Based on (Lamb and Roberts, 1998), in terms of the evolution operator, Eqn 4 implies:

$$R \circ \phi_t = \phi_{-t} \circ R = \phi_t^{-1} \circ R, \quad (5)$$

which means that moving forward $t$ steps and then turning to the opposite direction is equivalent to firstly turning to the opposite direction and then moving backwards $t$ steps[4]. Eqn 5 has been widely used to describe time-reversal symmetry in existing literature (Huh et al., 2020; Valperga et al., 2022). Nevertheless, we propose the following lemma, which is more intuitive to understand and straightforward to guide the design of our time-reversal regularizer.

**Lemma 2.1.** *Eqn 5 is equivalent to* $R \circ \phi_t \circ R \circ \phi_t = I$, *where $I$ denotes identity mapping.*

Lemma 2.1 means if we move $t$ steps forward, then turn to the opposite direction, and then move forward for $t$ more steps, it shall restore back to the same state. This is illustrated in Figure 2 where the reverse trajectory should be the same as the forward trajectory.[5] It can be understood as rewinding a video to the very beginning. The proof of Lemma 2.1 is in Appendix A.2.

## 3 Method: TREAT

We present a novel framework TREAT that achieves high-precision modeling for a wide range of systems from the numerical aspect, by enforcing Time-Reversal Symmetry (TRS) via a regularization

---

[4]Time-reversal symmetry is a property of physical systems, which requires the forward and reverse trajectories to be generated by the same mechanism $F(\cdot)$. It differs from reversibility of neural networks (Chang et al., 2018; Liu et al., 2019), which is a property of machine learning models and ensures the recovery of input from output via a reversed operator $f^{-1}(\cdot)$. We highlight the detailed discussions in Appendix F.

[5]We explain Figure 2 with implementation in Appendix A.1.

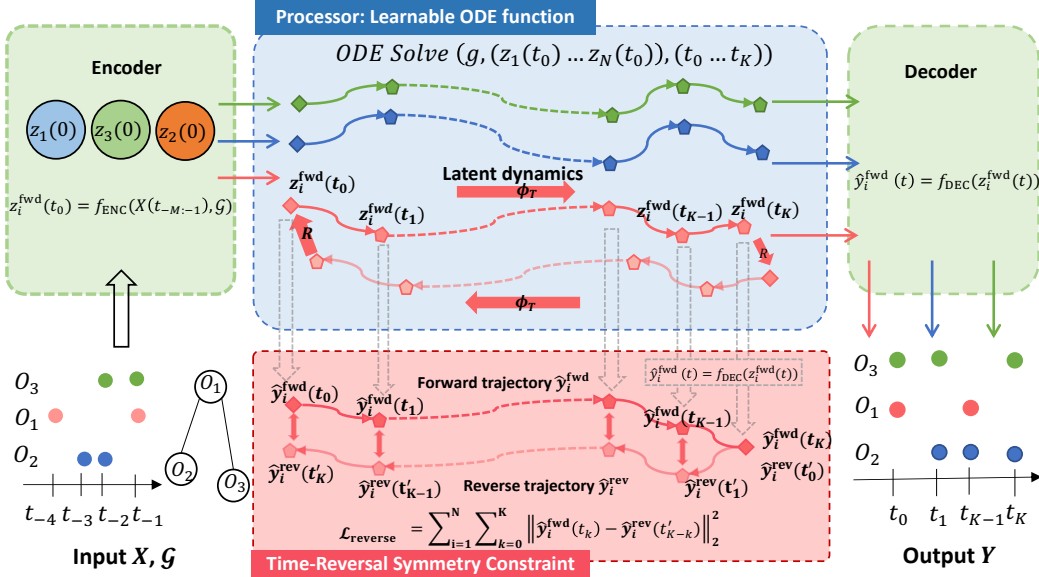

Figure 3: Overall framework of TREAT. $O_1, O_2, O_3$ are connected agents. It follows the encoder-processor-decoder architecture introduced in Sec 2.1. A novel TRS loss is incorporated to improve modeling accuracy across systems from the numerical aspect, regardless of their physical properties.

term. It improves modeling accuracy regardless of systems' physical properties. We first introduce our architecture design, followed by theoretical analysis to explain its numerical benefits.

TREAT uses GraphODE (Huang et al., 2020) as the backbone and flexibly incorporates TRS as a regularization term based on Lemma 2.1. This term aligns model forward and reverse trajectories. In practice, our model predicts the forward trajectories at a series of timestamps $\{t_k\}_{k=0}^{K}$ as ground truth observations are discrete, where $0 = t_0 < t_1 < \cdots < t_K = T$. The reverse trajectories are also at the same series of $K$ timestamps so as to be aligned with the forward one, which we denote as $\{t'_k\}_{k=0}^{K}$ satisfying $0 = t'_0 < t'_1 < \cdots < t'_K = T$. It's important to note that the values of the time variable $t'_k$ in the reverse trajectories do not represent real time, but serve as indexes of reverse trajectories. This leads to the relation $t'_{K-k} = T - t_k$, which means the reverse trajectories at timestamp $t'_{K-k}$ correspond to the forward trajectories at time $t_k$. For example, $t'_0 = T - t_K = 0$. It indicates $t'_0$ and $t_K$ are both pointing to the same real time $T$, which is the ending point of the forward trajectory as shown in Figure 3. Based on Lemma 2.1, the difference of the two trajectories at any observed time should be small, i.e. $\boldsymbol{z}^{\text{fwd}}(t_k) \approx \boldsymbol{z}^{\text{rev}}(t'_{K-k})$. This serves as the guideline for our regularizer design. The weight of the regularizer is also adjustable to adapt different systems. The overall framework is depicted in Figure 3.

### 3.1 Time-Reversal Symmetry Loss and Training

**Forward Trajectory Prediction and Reconstruction Loss.** For multi-agent systems, we utilize the GNN operator described in (Kipf et al., 2018) as our ODE function $g(\cdot)$, which drives the system to move forward and output the forward trajectories for latent states $\boldsymbol{z}_i^{\text{fwd}}(t)$ at each continuous time $t \in [0, T]$ and each agent $i$. We then employ a Multilayer Perceptron (MLP) as a decoder to predict output trajectories $\hat{\boldsymbol{y}}_i^{\text{fwd}}(t)$ based on the latent states. We summarize the whole procedure as:

$$
\begin{aligned}
\dot{\boldsymbol{z}}_i^{\text{fwd}}(t) &:= \frac{d\boldsymbol{z}_i^{\text{fwd}}(t)}{dt} = g(\boldsymbol{z}_1^{\text{fwd}}(t), \boldsymbol{z}_2^{\text{fwd}}(t), \cdots \boldsymbol{z}_N^{\text{fwd}}(t)), \\
\boldsymbol{z}_i^{\text{fwd}}(t_0) &= f_{\text{ENC}}(X(t_{-M:-1}), \mathcal{G}), \quad \hat{\boldsymbol{y}}_i^{\text{fwd}}(t) = f_{\text{DEC}}(\boldsymbol{z}_i^{\text{fwd}}(t)).
\end{aligned}
\tag{6}
$$

To train the model, we use the reconstruction loss that minimizes the L2 distance between predicted forward trajectories $\{\hat{\boldsymbol{y}}_i^{\text{fwd}}(t_k)\}_{k=0}^K$ and the ground truth trajectories $\{\boldsymbol{y}_i(t_k)\}_{k=0}^K$ as :

$$\mathcal{L}_{pred} = \sum_{i=1}^N \sum_{k=0}^K \left\| \boldsymbol{y}_i(t_k) - \hat{\boldsymbol{y}}_i^{\text{fwd}}(t_k) \right\|_2^2. \tag{7}$$

**Reverse Trajectory Prediction and Regularization Loss.** We design a novel time-reversal symmetry loss as a soft constraint to flexibly regulate systems' behavior based on Lemma 2.1. Specifically, we first compute the latent reverse trajectories $\boldsymbol{z}^{\text{rev}}(t)$ by starting from the ending state of the forward one, traversed back over time. We then employ the decoder to output dynamic trajectories $\boldsymbol{y}^{\text{rev}}(t)$.

$$\dot{\boldsymbol{z}}_i^{\text{rev}}(t) := \frac{d\boldsymbol{z}_i^{\text{rev}}(t)}{dt} = -g(\boldsymbol{z}_1^{\text{rev}}(t), \boldsymbol{z}_2^{\text{rev}}(t), \cdots \boldsymbol{z}_N^{\text{rev}}(t)),$$
$$\boldsymbol{z}_i^{\text{rev}}(t_0') = \boldsymbol{z}_i^{\text{fwd}}(t_K), \quad \hat{\boldsymbol{y}}_i^{\text{rev}}(t) = f_{\text{DEC}}(\boldsymbol{z}_i^{\text{rev}}(t)). \tag{8}$$

Next, based on Lemma 2.1, if the system follows *Time-Reversal Symmetry*, the forward and backward trajectories shall be exactly overlap. We thus design the reversal loss by minimizing the L2 distances between model forward and backward trajectories decoded from the latent trajectories:

$$\mathcal{L}_{reverse} = \sum_{i=1}^N \sum_{k=0}^K \left\| \hat{\boldsymbol{y}}_i^{\text{fwd}}(t_k) - \hat{\boldsymbol{y}}_i^{\text{rev}}(t_{K-k}') \right\|_2^2. \tag{9}$$

Finally, we jointly train TREAT as a weighted combination of the two losses:

$$\mathcal{L} = \mathcal{L}_{pred} + \alpha \mathcal{L}_{reverse} = \sum_{i=1}^N \sum_{k=0}^K \left\| \boldsymbol{y}_i(t_k) - \hat{\boldsymbol{y}}_i^{\text{fwd}}(t_k) \right\|_2^2 + \alpha \sum_{i=1}^N \sum_{k=0}^K \left\| \hat{\boldsymbol{y}}_i^{\text{fwd}}(t_k) - \hat{\boldsymbol{y}}_i^{\text{rev}}(t_{K-k}') \right\|_2^2, \tag{10}$$

where $\alpha$ is a positive coefficient to balance the two losses based on different targeted systems.

**Remark.** The computational time of $\mathcal{L}_{reverse}$ is of the same scale as the reconstruction loss $\mathcal{L}_{pred}$. As the computation process of the reversal loss is to first use the ODE solver to generate the reverse trajectories, which has the same computational overhead as computing the forward trajectories, and then compute the L2 distances.

## 3.2 Theoretical Analysis of Time-Reversal Symmetry Loss

We next theoretically show that the time-reversal symmetry loss numerically helps to improve prediction accuracy in general, regardless of systems' physical properties. Specifically, we show that it minimizes higher-order Taylor expansion terms during the ODE integration steps.

**Theorem 3.1.** *Let $\Delta t$ denote the integration step size in an ODE solver and $T$ be the prediction length. The reconstruction loss $\mathcal{L}_{pred}$ defined in Eqn 7 is $\mathcal{O}(T^3 \Delta t^2)$. The time-reversal loss $\mathcal{L}_{reverse}$ defined in Eqn 9 is $\mathcal{O}(T^5 \Delta t^4)$.*

We prove Theorem 3.1 in Appendix A.3. From Theorem 3.1, we can see two nice properties of our proposed time-reversal loss: 1) Regarding the relationship to $\Delta t$, $\mathcal{L}_{reverse}$ is optimizing a high-order term $\Delta t^4$, which forces the model to predict fine-grained physical properties such as jerk (the derivatives of accelerations). In comparison, the reconstruction loss optimizes $\Delta t^2$, which mainly guides the model to predict the locations/velocities accurately. Therefore, the combined loss enables our model to be more noise-tolerable; 2) Regarding the relationship to $T$, $\mathcal{L}_{reverse}$ is more sensitive to total sequence length ($T^5$), thus it provides more regularization for long-context prediction, a key challenge for dynamic modeling.

**TRS Loss Design Choice.** We define $\mathcal{L}_{reverse}$ as the distance between model forward trajectories and backward trajectories. Based on the definition of TRS in Sec. 2.2, there are other implementation choices. One prior work TRS-ODE (Huh et al., 2020) designed a TRS loss based on Eqn 5, where a reverse trajectory shares the same starting point as the forward one. However, we show that our implementation based on Lemma 2.1 to approximate time-reversal symmetry has a lower maximum error compared to their implementation below, supported by empirical experiments in Sec. 4.2.

**Lemma 3.2.** *Let $\mathcal{L}_{reverse}$ be the TRS implementation of TREAT based on Lemma 2.1, $\mathcal{L}_{reverse2}$ be the one in (Huh et al., 2020) based on Eqn 5. When the reconstruction loss defined in Eqn 7 of both methods are equal, and the two TRS losses are equal, i.e. $\mathcal{L}_{reverse} = \mathcal{L}_{reverse2}$, the maximum error between the reversal and ground truth trajectory for each agent, i.e. $MaxError_{gt\_rev} = \max_{k \in [K]} \|\boldsymbol{y}_i(t_k) - \hat{\boldsymbol{y}}_i^{\text{rev}}(t'_{K-k})\|_2$ for $i = 1, 2 \cdots N$, made by TREAT is smaller.*

We prove Lemma 3.2 in Appendix A.4. Another implementation is to minimize the distances between model backward trajectories and ground truth trajectories. When both forward and backward trajectories are close to ground-truth, they are implicitly symmetric. The major drawback is that at the early stage of learning when the forward is far away from ground truth ($\mathcal{L}_{pred}$), such implicit regularization does not force time-reversal symmetry, but introduces more noise.

# 4    Experiments

**Datasets.** We conduct systematic evaluations over five multi-agent systems including three 5-body spring systems (Kipf et al., 2018), a complex chaotic pendulum system and a real-world motion capture dataset (Carnegie Mellon University, 2003); and four single-agent systems including three spring systems (with only one node) and a chaotic strange attractors system  (Huh et al., 2020).

The settings of spring systems include: 1) conservative, i.e. no interactions with the environments, we call it *Simple Spring*; 2) non-conservative with frictions, we call it *Damped Spring*; 3) non-conservative with periodic external forces, we call it *Forced Spring*. The *Pendulum* system contains three connected sticks in a 2D plane. It is highly sensitive to initial states, with minor disturbances leading to significantly different trajectories (Shinbrot et al., 1992; Awrejcewicz et al., 2008). The real-world motion capture dataset  (Carnegie Mellon University, 2003) describes the walking trajectories of a person, each tracking a single joint. We call it *Human Motion*. The strange attractor consists of symmetric attractor/repellor force pairs and is chaotic (Sprott, 2015). It is also highly sensitive to the initial states (Koppe et al., 2019). We call it *Attractor*.

Towards physical properties, *Simple Spring* and *Pendulum* are conservative and reversible; *Force Spring* and *Attractor* are reversible but non-conservative; *Damped Spring* are irreversible and non-conservative. For *Human Motion*, it does not adhere to specific physical laws since it is a real-world dataset. Details of the datasets and generation pipelines can be found inAppendix C.

**Task Setup.** We conduct evaluation by splitting trajectories into two halves: $[t_1, t_M], [t_{M+1}, t_K]$ where timestamps can be irregular. We condition the first half of observations to make predictions for the second half as in (Rubanova et al., 2019). For spring datasets and *Pendulum*, we generate irregular-sampled trajectories and set the training samples to be 20,000 and testing samples to be 5,000 respectively. For *Attractor*, We generate 1,000 and 50 trajectories for training and testing respectively following Huh et al. (2020). 10% of training samples are used as validation sets and the maximum trajectory prediction length is 60. Details can be found in Appendix C.

**Baselines.** We compare TREAT against three baseline types: 1) pure data-driven approaches including LG-ODE (Huang et al., 2020) and LatentODE (Rubanova et al., 2019), where the first one is a multi-agent approach considering pair-wise interactions, and the second one is a single-agent approach that predicts each trajectory independently; 2) energy-preserving HODEN (Greydanus et al., 2019); and 3) time-reversal TRS-ODEN (Huh et al., 2020).

The latter two are single-agent approaches and require initial states as given input. To handle missing initial states in our dataset, we approximate the initial states for the two methods via linear spline interpolation (Endre Süli, 2003). In addition, we substitute the ODE network in TRS-ODEN with a GNN (Kipf et al., 2018) as TRS-ODEN$_{\text{GNN}}$, which serves as a new multi-agent approach for fair comparison. HODEN cannot be easily extended to the multi-agent setting as replacing the ODE function with a GNN can violate energy conservation of the original HODEN. For running LGODE and TREAT on single-agent datasets, we only include self-loop edges in the graph $\mathcal{G} = (\mathcal{V}, \mathcal{E})$, which makes the ODE function $g$ a simple MLP. Implementation details can be found in Appendix D.2.

Table 1: Evaluation results on MSE ($10^{-2}$). Best results are in **bold** numbers and second-best results are in underline numbers. *Human Motion* is a real-world dataset and all others are simulated datasets.

| Dataset | Multi-Agent Systems | | | | | Single-Agent Systems | | | |
|---|---|---|---|---|---|---|---|---|---|
| | *Simple Spring* | *Forced Spring* | *Damped Spring* | *Pendulum* | *Human Motion* | *Simple Spring* | *Forced Spring* | *Damped Spring* | *Attractor* |
| LatentODE | 5.2622 | 5.0277 | 3.3419 | 2.6894 | 2.9061 | 5.7957 | 0.4563 | 1.3012 | 0.58394 |
| HODEN | 3.0039 | 4.0668 | 8.7950 | 741.2296 | 1.9855 | 3.2119 | 4.004 | 1.5675 | 54.2912 |
| TRS-ODEN | 3.6785 | 4.4465 | 1.7595 | 741.4988 | 0.5400 | 3.0271 | 0.4056 | 1.5667 | 2.2683 |
| TRS-ODEN$_{GNN}$ | 1.4115 | 2.1102 | 0.5951 | 596.0319 | 0.2609 | / | / | / | / |
| LG-ODE | 1.7429 | 1.8929 | 0.9718 | 1.4156 | 0.7610 | 1.6156 | 0.1465 | 1.1223 | 0.6942 |
| TREAT | **1.1178** | **1.4525** | **0.5944** | **1.2527** | **0.2192** | **1.6026** | **0.0960** | **1.0750** | **0.5581** |
| (—-Ablation of our method with different implementation of $L_{reverse}$—-) | | | | | | | | | |
| TREAT$_{\mathcal{L}_{rev}=\text{gt-rev}}$ | 1.1313 | 1.5254 | 0.6171 | 1.6158 | 0.2495 | 1.6190 | 0.1104 | 1.1205 | 0.6364 |
| TREAT$_{\mathcal{L}_{rev}=\text{rev2}}$ | 1.6786 | 1.9786 | 0.9692 | 1.5631 | 0.8785 | 1.6901 | 0.0983 | 1.0952 | 0.7286 |

## 4.1 Main Results

Table 1 shows the prediction performance on both multi-agent systems and single-agent systems measured by mean squared error (MSE). We can see that TREAT consistently surpasses other models, highlighting its generalizability and the efficacy of the proposed TRS loss.

For multi-agent systems, approaches that consider interactions among agents (LG-ODE, TRS-ODEN$_{GNN}$, TREAT) consistently outperform single-agent baselines (LatentODE, HODEN, TRS-ODEN), and TREAT achieves the best performance across datasets.

The chaotic nature of the *Pendulum* system and the *Attractor* system, with their sensitivity to initial states [6], poses extreme challenges for dynamic modeling. This leads to highly unstable predictions for models like HODEN and TRS-ODEN, as they estimate initial states via inaccurate linear spline interpolation (Endre Süli, 2003). In contrast, LatentODE, LG-ODE, and TREAT employ advanced encoders that infer latent states from observed data and demonstrate superior accuracy. Among them, TREAT achieves the most accurate predictions, further showing its robust generalization capabilities.

We observe that misapplied inductive biases can degrade results, which limits the usage of physics-informed methods that are designed for individual physical prior such as HODEN. HODEN only excels on energy-conservative systems, such as *Simple Spring* compared with LatentODE and TRS-ODEN in the multi-agent setting. Its performance drop dramatically on *Force Spring*, *Damped Spring*, and *Attractor*. Note that HODEN naively forces each agent to be energy-conservative, instead of the whole system. Therefore, it performs poorly than LG-ODE, TREAT in the multi-agent settings.

For the *Human Motion* dataset, characterized by its dynamic ambiguity as it does not adhere to specific physical laws, we cannot directly determine whether it is conservative or time-reversal. For such a system with an unknown nature, TREAT outperforms other purely data-driven methods significantly, showcasing its strong numerical benefits in improving prediction accuracy across diverse system types. This is also shown by its superior performance on *Damped Spring*, which is irreversible.

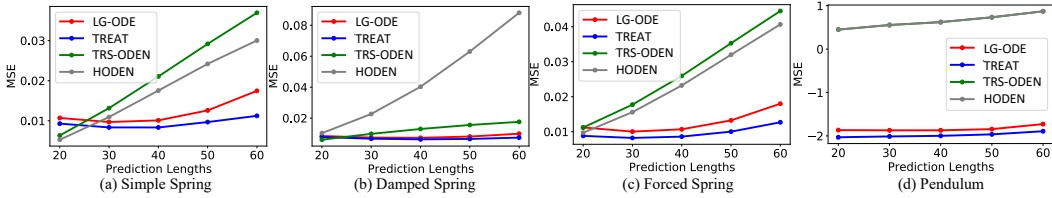

Figure 4: Varying prediction lengths across multi-agent datasets (Pendulum MSE is in log values).

---

[6]Video to show *Pendulum* is highly sensitive to initial states.

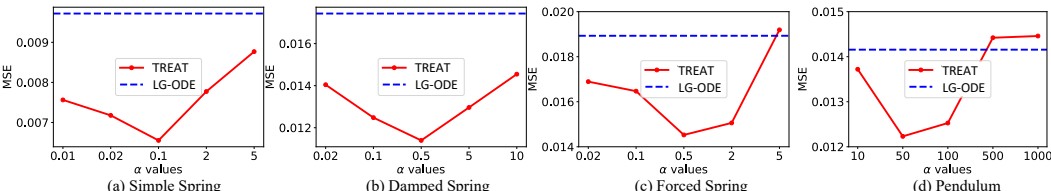

Figure 5: Varying $\alpha$ values across multi-agent datasets.

## 4.2 Ablation and Sensitivity Analysis

**Ablation on implementation of $\mathcal{L}_{reverse}$.** We conduct two ablation by changing the implementation of $\mathcal{L}_{reverse}$ discussed in Sec. 3.2: 1) TREAT$_{\mathcal{L}_{rev}=\text{gt-rev}}$ , which computes the reversal loss as the L2 distance between ground truth trajectories to model backward trajectories; 2) TREAT$_{\mathcal{L}_{rev}=\text{rev2}}$, which implements the TRS loss based on Eqn 5 as in TRS-ODEN (Huh et al., 2020). From the last block of Table 1, we can clearly see that our implementation achieves the best performance against the two.

**Evaluation across prediction lengths.** We vary the maximum prediction lengths from 20 to 60 and report model performance as shown in Figure 4. As the prediction step increases, TREAT consistently maintains optimal prediction performance, while other baselines exhibit significant error accumulations. The performance gap between TREAT and baselines widens when making long-range predictions, highlighting the superior predictive capability of TREAT.

**Evaluation across different $\alpha$.** We vary the values of the coefficient $\alpha$ defined in Eqn 10, which balances the reconstruction loss and the TRS loss. Figure 5 demonstrates that the optimal $\alpha$ values being neither too high nor too low. This is because when $\alpha$ is too small, the model tends to neglect the TRS physical bias, resulting in error accumulations. Conversely, when $\alpha$ becomes too large, the model can emphasize TRS at the cost of accuracy. Nonetheless, across different $\alpha$ values, TREAT consistently surpasses the purely data-driven LG-ODE, showcasing its superiority and flexibility in modeling diverse dynamical systems.

We study TREAT's sensitivity towards solver choice and observation ratios in Appendix E.1 and Appendix E.2 respectively.

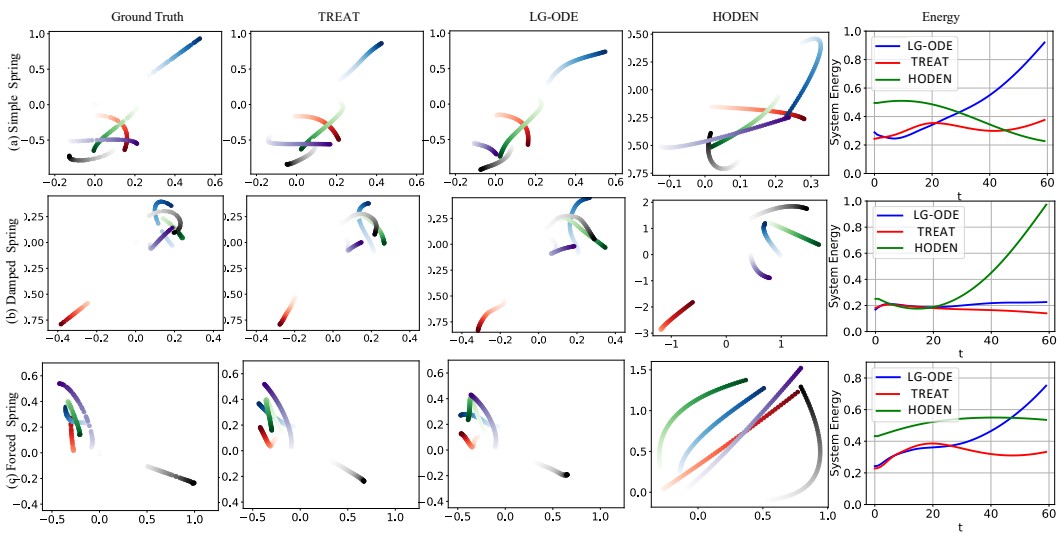

Figure 6: Visualization for 5-body spring systems (trajectory starts from light to dark colors).

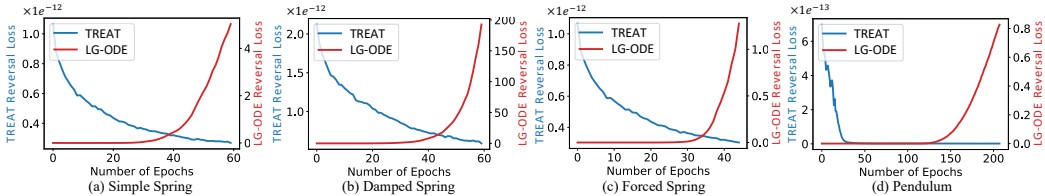

Figure 7: TRS loss visualization across multi-agent datasets (scales of two y-axes are different).

## 4.3 Visualizations

**Trajectory Visualizations.** Model predictions and ground truth are visualized in Figure 6. As HODEN is a single-agent baseline that individually forces every agent's energy to be constant over time which is not valid, the predicted trajectories is having the largest errors and systems' total energy is not conserved for all datasets. The purely data-driven LG-ODE exhibits unrealistic energy patterns, as seen in the energy spikes in *Simple Spring* and *Force Spring*. In contrast, TREAT, incorporating reversal loss, generates realistic energy trends, and consistently produces trajectories closest to the ground truth, showing its superior performance.

**Reversal Loss Visualizations** To illustrate the issue of energy explosion from the purely data-driven LG-ODE, we visualize the TRS loss over training epochs from LG-ODE[7] and TREAT in Figure 7. As results suggest, LG-ODE has increased TRS loss over training epochs, meaning it is violating the time-reversal symmetry sharply, in contrast to TREAT which has decreased reversal loss over epochs.

## 5 Conclusions

We propose TREAT, a deep learningframework that achieves high-precision modeling for a wide range of dynamical systems by injecting time-reversal symmetry as an inductive bias. TREAT features a novel regularization term to softly enforce time-reversal symmetry by aligning predicted forward and reverse trajectories from a GraphODE model. Notably, we theoretically prove that the regularization term effectively minimizes higher-order Taylor expansion terms during the ODE integration, which serves as a general numerical benefit widely applicable to various systems (even irreversible systems) regardless of their physical properties. Empirical evaluations on different kinds of datasets illustrate TREAT's superior efficacy in accurately capturing real-world system dynamics.

## 6 Limitations

Currently, TREAT only incorporates inductive bias from the temporal aspect, while there are many important properties in the spatial aspect such as translation and rotation equivariance (Satorras et al., 2021; Han et al., 2022b; Xu et al., 2022). Future endeavors that combine biases from both temporal and spatial dimensions could unveil a new frontier in dynamical systems modeling.

## 7 Acknowledgement

This work was partially supported by NSF 2200274, NSF 2106859, NSF 2312501, NSF 2211557, NSF 1937599, NSF 2119643, NSF 2303037, NSF 2312501, DARPA HR00112290103/HR0011260656, HR00112490370, NIH U54HG012517, NIH U24DK097771, NASA, SRC JUMP 2.0 Center, Amazon Research Awards, and Snapchat Gifts.

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

# A Theoretical Analysis

## A.1 Implementation of the Time-Reversal Symmetry Loss

---
**Algorithm 1** The implementation of $\mathcal{L}_{reverse}$

---
**Require:** latent initial states $z_i^{\text{fwd}}(t_0)$; the ODE function $g(\cdot)$; number of agents $N$:
 1: **for** each $i \in N$ **do**
 2:     Compute the latent forward trajectory at timestamps $\{t_k\}_{k=0}^{K}$:
     $z_i^{\text{fwd}}(t_k) = \text{ODE-Solver}\big(g, [z_1^{\text{fwd}}(t_0), z_2^{\text{fwd}}(t_0)...z_N^{\text{fwd}}(t_0)], t_k\big)$. Reach the final state $z_i^{\text{fwd}}(t_K)$.

 3:     The initial state of the reverse trajectory is defined as $z_i^{\text{rev}}(t_0') = z_i^{\text{fwd}}(t_K)$, and the dynamics
     of the system which is the ODE function $g(\cdot)$ is also reversed as $-g(\cdot)$ .
 4:     Compute the latent reverse trajectory at timestamps $\{t_k'\}_{k=0}^{K}$,
     $z_i^{\text{rev}}(t_k') = \text{ODE-Solver}\big(g, [z_1^{\text{rev}}(t_0'), z_2^{\text{rev}}(t_0')...z_N^{\text{rev}}(t_0')], t_k'\big)$.
 5:     $\hat{y}_i^{\text{fwd}}(t_k) = f_{\text{DEC}}(z_i^{\text{fwd}}(t_k))$ ,$\hat{y}_i^{\text{rev}}(t_k') = f_{\text{DEC}}(z_i^{\text{rev}}(t_k'))$
 6: **end for**
 7: $\mathcal{L}_{reverse} = \sum_{i=1}^{N} \sum_{k=0}^{K} \left\| \hat{y}_i^{\text{fwd}}(t_k) - \hat{y}_i^{\text{rev}}(t_{K-k}') \right\|_2^2$

---

## A.2 Proof of Lemma 1

*Proof.* The definition of time-reversal symmetry is given by:

$$R \circ \phi_t = \phi_{-t} \circ R = \phi_t^{-1} \circ R \tag{11}$$

Here, $R$ is an involution operator, which means $R \circ R = \text{I}$.

First, we apply the time evolution operator $\phi_t$ to both sides of Eqn 11:

$$\phi_t \circ R \circ \phi_t = \phi_t \circ \phi_t^{-1} \circ R \tag{12}$$

Simplifying, we obtain:

$$\phi_t \circ R \circ \phi_t = R \tag{13}$$

Next, we apply the involution operator $R$ to both sides of the equation:

$$R \circ \phi_t \circ R \circ \phi_t = R \circ R \tag{14}$$

Since $R \circ R = \text{I}$, we finally arrive at:

$$R \circ \phi_t \circ R \circ \phi_t = \text{I} \tag{15}$$

which means the trajectories can overlap when evolving backward from the final state. $\qquad\square$

## A.3 Proof of Theorem 3.1

Let $\Delta t$ denote the integration step size in an ODE solver and $T$ be the prediction length. The time stamps of the ODE solver are $\{t_j\}_{j=0}^{T}$, where $t_{j+1} - t_j = \Delta t$ for $j = 0, \cdots, T(T > 1)$. Next suppose during the forward evolution, the updates go through states $z^{\text{fwd}}(t_j) = (q^{\text{fwd}}(t_j), p^{\text{fwd}}(t_j))$ for $j = 0, \cdots, T$, where $q^{\text{fwd}}(t_j)$ is position, $p^{\text{fwd}}(t_j)$ is momentum, while during the reverse evolution they go through states $z^{\text{rev}}(t_j) = (q^{\text{rev}}(t_j), p^{\text{rev}}(t_j))$ for $j = 0, \cdots, T$, in reverse order. The ground truth trajectory is $z^{\text{gt}}(t_j) = (q^{\text{gt}}(t_j), p^{\text{gt}}(t_j))$ for $j = 0, \cdots, T$.

For the sake of brevity in the ensuing proof, we denote $z^{\text{gt}}(t_j)$ by $z_j^{\text{gt}}$, $z^{\text{fwd}}(t_j)$ by $z_j^{\text{fwd}}$ and $z^{\text{rev}}(t_j)$ by $z_j^{\text{rev}}$, and we will use Mathematical Induction to prove the theorem.

### A.3.1 Reconstruction Loss ($\mathcal{L}_{pred}$) Analysis.

First, we bound the forward loss $\sum_{j=0}^{T} \|z_j^{\text{fwd}} - z_j^{\text{gt}}\|_2^2$. Since our method models the momentum and position of the system, we can write the following Taylor expansion of the forward process, where

for any $0 \le j < T$:

$$
\begin{cases}
\boldsymbol{q}_{j+1}^{\text{fwd}} = \boldsymbol{q}_j^{\text{fwd}} + (\boldsymbol{p}_j^{\text{fwd}}/m)\Delta t + (\dot{\boldsymbol{p}}_j^{\text{fwd}}/2m)\Delta t^2 + \mathcal{O}(\Delta t^3), & \text{(16a)} \\
\boldsymbol{p}_{j+1}^{\text{fwd}} = \boldsymbol{p}_j^{\text{fwd}} + \dot{\boldsymbol{p}}_j^{\text{fwd}}\Delta t + \mathcal{O}(\Delta t^2), & \text{(16b)} \\
\dot{\boldsymbol{p}}_{j+1}^{\text{fwd}} = \dot{\boldsymbol{p}}_j^{\text{fwd}} + \mathcal{O}(\Delta t), & \text{(16c)}
\end{cases}
$$

and for the ground truth process, we also have from Taylor expansion that

$$
\begin{cases}
\boldsymbol{q}_{j+1}^{\text{gt}} = \boldsymbol{q}_j^{\text{gt}} + (\boldsymbol{p}_j^{\text{gt}}/m)\Delta t + (\dot{\boldsymbol{p}}_j^{\text{gt}}/2m)\Delta t^2 + \mathcal{O}(\Delta t^3), & \text{(17a)} \\
\boldsymbol{p}_{j+1}^{\text{gt}} = \boldsymbol{p}_j^{\text{gt}} + \dot{\boldsymbol{p}}_j^{\text{gt}}\Delta t + \mathcal{O}(\Delta t^2), & \text{(17b)} \\
\dot{\boldsymbol{p}}_{j+1}^{\text{gt}} = \dot{\boldsymbol{p}}_j^{\text{gt}} + \mathcal{O}(\Delta t). & \text{(17c)}
\end{cases}
$$

With these, we aim to prove that for any $k = 0, 1, \cdots, T$, the following hold :

$$
\begin{cases}
\|\boldsymbol{q}_k^{\text{fwd}} - \boldsymbol{q}_k^{\text{gt}}\|_2 \le C_2^{\text{fwd}} k^2 \Delta t^2, & \text{(18a)} \\
\|\boldsymbol{p}_k^{\text{fwd}} - \boldsymbol{p}_k^{\text{gt}}\|_2 \le C_1^{\text{fwd}} k \Delta t, & \text{(18b)}
\end{cases}
$$

where $C_1^{\text{fwd}}$ and $C_2^{\text{fwd}}$ are constants.

**Base Case** $k = 0$**:** Based on the initialization rules, it is obvious that $\left\|\boldsymbol{q}_0^{\text{fwd}} - \boldsymbol{q}_0^{\text{gt}}\right\|_2 = 0$ and $\left\|\boldsymbol{p}_0^{\text{fwd}} - \boldsymbol{p}_0^{\text{gt}}\right\|_2 = 0$, thus (18a) and (18b) both hold for $k = 0$.

**Inductive Hypothesis:** Assume (18a) and (18b) hold for $k = j$, which means:

$$
\begin{cases}
\|\boldsymbol{q}_j^{\text{fwd}} - \boldsymbol{q}_j^{\text{gt}}\|_2 \le C_2^{\text{fwd}} j^2 \Delta t^2, & \text{(19a)} \\
\|\boldsymbol{p}_j^{\text{fwd}} - \boldsymbol{p}_j^{\text{gt}}\|_2 \le C_1^{\text{fwd}} j \Delta t, & \text{(19b)}
\end{cases}
$$

**Inductive Proof:** We need to prove (18a) and (18b) hold for $k = j + 1$.

First, using (16c) and (17c), we have

$$
\left\|\dot{\boldsymbol{p}}_{j+1}^{\text{fwd}} - \dot{\boldsymbol{p}}_{j+1}^{\text{gt}}\right\|_2 = \left\|\dot{\boldsymbol{p}}_j^{\text{fwd}} - \dot{\boldsymbol{p}}_j^{\text{gt}}\right\|_2 + \mathcal{O}(\Delta t) = \left\|\dot{\boldsymbol{p}}_0^{\text{fwd}} - \dot{\boldsymbol{p}}_0^{\text{gt}}\right\|_2 + \mathcal{O}\big((j+1)\Delta t\big) = \mathcal{O}(1), \quad (20)
$$

where we iterate through $j, j - 1, \cdots, 0$ in the second equality. Then using (17b) and (16b), we get for $j + 1$ that

$$
\begin{aligned}
\left\|\boldsymbol{p}_{j+1}^{\text{fwd}} - \boldsymbol{p}_{j+1}^{\text{gt}}\right\|_2 &= \left\|\big(\boldsymbol{p}_j^{\text{fwd}} + \dot{\boldsymbol{p}}_j^{\text{fwd}}\Delta t\big) - \big(\boldsymbol{p}_j^{\text{gt}} + \dot{\boldsymbol{p}}_j^{\text{gt}}\Delta t\big) + \mathcal{O}(\Delta t^2)\right\|_2 \\
&\le \left\|\boldsymbol{p}_j^{\text{fwd}} - \boldsymbol{p}_j^{\text{gt}}\right\|_2 + \left\|\dot{\boldsymbol{p}}_j^{\text{fwd}} - \dot{\boldsymbol{p}}_j^{\text{gt}}\right\|_2 \Delta t + \mathcal{O}(\Delta t^2) \\
&\le \left[C_1^{\text{fwd}} j + \mathcal{O}(1)\right]\Delta t,
\end{aligned}
$$

where the first inequality uses the triangle inequality, and in the second inequality we use (19b) as well as (20). We can see there exists $C_1^{\text{fwd}}$ such that the final expression above is upper bounded by $C_1^{\text{fwd}}(j + 1)\Delta t$, with which the claim holds for $j + 1$.

Next for (18a), using (17a) and (16a), we get for any $j$ that

$$
\begin{aligned}
\left\|\boldsymbol{q}_{j+1}^{\text{fwd}} - \boldsymbol{q}_{j+1}^{\text{gt}}\right\|_2 &= \left\|\big(\boldsymbol{q}_j^{\text{fwd}} + (\boldsymbol{p}_j^{\text{fwd}}/m)\Delta t + (\dot{\boldsymbol{p}}_j^{\text{fwd}}/2m)\Delta t^2\big) - \big(\boldsymbol{q}_j^{\text{gt}} + (\boldsymbol{p}_j^{\text{gt}}/m)\Delta t + (\dot{\boldsymbol{p}}_j^{\text{gt}}/2m)\Delta t^2\big) + \mathcal{O}(\Delta t^3)\right\|_2 \\
&\le \left\|\boldsymbol{q}_j^{\text{fwd}} - \boldsymbol{q}_j^{\text{gt}}\right\|_2 + \frac{1}{m}\left\|\boldsymbol{p}_j^{\text{fwd}} - \boldsymbol{p}_j^{\text{gt}}\right\|_2 \Delta t + \frac{1}{2m}\left\|\dot{\boldsymbol{p}}_j^{\text{fwd}} - \dot{\boldsymbol{p}}_j^{\text{gt}}\right\|_2 \Delta t^2 + \mathcal{O}(\Delta t^3) \\
&\le \left[C_2^{\text{fwd}} j^2 + \frac{C_1^{\text{fwd}}}{m} j + \mathcal{O}(1)\right]\Delta t^2,
\end{aligned}
$$

where the first inequality uses the triangle inequality, and in the second inequality we use (19a) and (19b) as well as (20). Thus with an appropriate $C_2^{\text{fwd}}$, we have the final expression above is upper bounded by $C_2^{\text{fwd}}(j + 1)^2 \Delta t^2$, and so the claim holds for $j + 1$.

Since both the base case and the inductive step have been proven, by the principle of mathematical induction, (18a) and (18b) holds for all $k = 0, 1, \cdots, T$.

With this, we finish the forward proof by plugging (18a) and (18b) into the loss function:

$$\sum_{j=0}^{T} \|z_j^{\text{fwd}} - z_j^{\text{gt}}\|_2^2 = \sum_{j=0}^{T} \|p_j^{\text{fwd}} - p_j^{\text{gt}}\|_2^2 + \sum_{j=0}^{T} \|q_j^{\text{fwd}} - q_j^{\text{gt}}\|_2^2$$

$$\leq \left(C_1^{\text{fwd}}\right)^2 \sum_{j=0}^{T} j^2 \Delta t^2 + \left(C_2^{\text{fwd}}\right)^2 \sum_{j=0}^{T} j^4 \Delta t^4$$

$$= \mathcal{O}(T^3 \Delta t^2).$$

### A.3.2   Reversal Loss ($\mathcal{L}_{reverse}$) Analysis.

Next we analyze the reversal loss $\sum_{j=0}^{T} \|R(z_j^{\text{rev}}) - z_j^{\text{fwd}}\|_2^2$. For this, we need to refine the Taylor expansion residual terms for a more in-depth analysis.

First reconsider the forward process. Since the process is generated from the learned network, we may assume that for some constants $c_1$, $c_2$, and $c_3$, the states satisfy the following for any $0 \leq j < T$:

$$\begin{cases} q_j^{\text{fwd}} = q_{j+1}^{\text{fwd}} - (p_{j+1}^{\text{fwd}}/m)\Delta t + (\dot{p}_{j+1}^{\text{fwd}}/2m)\Delta t^2 + \mathbf{rem}_j^{\text{fwd},3}, & \text{(21a)} \\ p_j^{\text{fwd}} = p_{j+1}^{\text{fwd}} - \dot{p}_{j+1}^{\text{fwd}}\Delta t + \mathbf{rem}_j^{\text{fwd},2}, & \text{(21b)} \\ \dot{p}_j^{\text{fwd}} = \dot{p}_{j+1}^{\text{fwd}} + \mathbf{rem}_j^{\text{fwd},1}, & \text{(21c)} \end{cases}$$

where the remaining terms $\left\|\mathbf{rem}_j^{\text{fwd},i}\right\|_2 \leq c_i \Delta t^i$ for $i = 1, 2, 3$. Similarly, we have approximate Taylor expansions for the reverse process:

$$\begin{cases} q_j^{\text{rev}} = q_{j+1}^{\text{rev}} + (p_{j+1}^{\text{rev}}/m)\Delta t + (\dot{p}_{j+1}^{\text{rev}}/2m)\Delta t^2 + \mathbf{rem}_j^{\text{rev},3}, & \text{(22a)} \\ p_j^{\text{rev}} = p_{j+1}^{\text{rev}} + \dot{p}_{j+1}^{\text{rev}}\Delta t + \mathbf{rem}_j^{\text{rev},2}, & \text{(22b)} \\ \dot{p}_j^{\text{rev}} = \dot{p}_{j+1}^{\text{rev}} + \mathbf{rem}_j^{\text{rev},1}, & \text{(22c)} \end{cases}$$

where $\left\|\mathbf{rem}_j^{\text{rev},i}\right\|_2 \leq c_i \Delta t^i$ for $i = 1, 2, 3$.

We will prove via induction that for $k = T, T - 1, \cdots, 0$,

$$\begin{cases} \|R(q_k^{\text{rev}}) - q_k^{\text{fwd}}\|_2 \leq C_3^{\text{rev}}(T-k)^3 \Delta t^3, & \text{(23a)} \\ \|R(p_k^{\text{rev}}) - p_k^{\text{fwd}}\|_2 \leq C_2^{\text{rev}}(T-k)^2 \Delta t^2, & \text{(23b)} \\ \|R(\dot{p}_k^{\text{rev}}) - \dot{p}_k^{\text{fwd}}\|_2 \leq C_1^{\text{rev}}(T-k)\Delta t, & \text{(23c)} \end{cases}$$

where $C_1^{\text{rev}}$, $C_2^{\text{rev}}$ and $C_3^{\text{rev}}$ are constants.

The entire proof process is analogous to the previous analysis of Reconstruction Loss.

**Base Case $k = T$:** Since the reverse process is initialized by the forward process variables at $k = T$, it is obvious that $\left\|q_T^{\text{fwd}} - q_T^{ev}\right\|_2 = \left\|p_T^{\text{fwd}} - p_T^{\text{rev}}\right\|_2 = \left\|\dot{p}_T^{\text{fwd}} - \dot{p}_T^{\text{rev}}\right\|_2 = 0$. Thus (23a), (23b) and (23c) all hold for $k = 0$.

**Inductive Hypothesis:** Assume the inequalities (23b), (23a) and (23c) hold for $k = j + 1$, which means:

$$\begin{cases} \|R(q_{j+1}^{\text{rev}}) - q_{j+1}^{\text{fwd}}\|_2 \leq C_3^{\text{rev}}(T-(j+1))^3 \Delta t^3, & \text{(24a)} \\ \|R(p_{j+1}^{\text{rev}}) - p_{j+1}^{\text{fwd}}\|_2 \leq C_2^{\text{rev}}(T-(j+1))^2 \Delta t^2, & \text{(24b)} \\ \|R(\dot{p}_{j+1}^{\text{rev}}) - \dot{p}_{j+1}^{\text{fwd}}\|_2 \leq C_1^{\text{rev}}(T-(j+1))\Delta t, & \text{(24c)} \end{cases}$$

**Inductive Proof:** We need to prove (23b) (23a) and (23c) holds for $k = j$.

First, for (23c), using (21c) and (22c), we get for any $j$ that

$$\left\|R(\dot{p}_j^{\text{rev}}) - \dot{p}_j^{\text{fwd}}\right\|_2$$
$$= \left\|(\dot{p}_{j+1}^{\text{rev}} + \mathbf{rem}_j^{\text{rev},1}) - (\dot{p}_{j+1}^{\text{fwd}} + \mathbf{rem}_j^{\text{fwd},1})\right\|_2$$
$$\leq \left\|R(\dot{p}_{j+1}^{\text{rev}}) - \dot{p}_{j+1}^{\text{fwd}}\right\|_2 + \|\mathbf{rem}_j^{\text{rev},1}\|_2 + \|\mathbf{rem}_j^{\text{fwd},1}\|_2$$
$$\leq C_1^{\text{rev}}(T - j - 1)\Delta t + 2c_1 \Delta t,$$

where the first inequality uses the triangle inequality, and the second inequality plugs in (24c). Thus taking $C_1^{\text{rev}} = 2c_1$, the above is upped bounded by $C_1^{\text{rev}}(T-j)\Delta t$, and (23b) holds for $j$.

Second, for (24b), using (21b) and (22b), we get

$$
\begin{aligned}
\left\| R(\boldsymbol{p}_j^{\text{rev}}) - \boldsymbol{p}_j^{\text{fwd}} \right\|_2 &= \left\| -(\boldsymbol{p}_{j+1}^{\text{rev}} + \dot{\boldsymbol{p}}_{j+1}^{\text{rev}}\Delta t + \mathbf{rem}_j^{\text{rev},2}) - (\boldsymbol{p}_{j+1}^{\text{fwd}} - \dot{\boldsymbol{p}}_{j+1}^{\text{fwd}}\Delta t + \mathbf{rem}_j^{\text{fwd},2}) \right\|_2 \\
&\le \left\| R(\boldsymbol{p}_{j+1}^{\text{rev}}) - \boldsymbol{p}_{j+1}^{\text{fwd}} \right\|_2 + \left\| R(\dot{\boldsymbol{p}}_{j+1}^{\text{rev}}) - \dot{\boldsymbol{p}}_{j+1}^{\text{fwd}} \right\|_2\Delta t + \|\mathbf{rem}_j^{\text{rev},2}\|_2 + \|\mathbf{rem}_j^{\text{fwd},2}\|_2 \\
&\le \left[ C_2^{\text{rev}}(T-j-1)^2 + C_1^{\text{rev}}(T-j-1) + 2c_2 \right]\Delta t^2,
\end{aligned}
$$

where the first inequality uses the triangle inequality, and in the second inequality we use (24a) and (24b). Thus taking $C_2^{\text{rev}} = \max\{C_1^{\text{rev}}/2, 2c_2\}$, we have the final expression above is upper bounded by $C_2^{\text{rev}}(T-j)^2\Delta t^2$, and so the claim holds for $j$.

Finally, for (24a), we use (21a) and (22a) to get

$$
\begin{aligned}
&\left\| R(\boldsymbol{q}_j^{\text{rev}}) - \boldsymbol{q}_j^{\text{fwd}} \right\|_2 \\
&= \left\| (\boldsymbol{q}_{j+1}^{\text{rev}} + (\boldsymbol{p}_{j+1}^{\text{rev}}/m)\Delta t + (\dot{\boldsymbol{p}}_{j+1}^{\text{rev}}/2m)\Delta t^2 + \mathbf{rem}_j^{\text{rev},3}) - (\boldsymbol{q}_{j+1}^{\text{fwd}} - (\boldsymbol{p}_{j+1}^{\text{fwd}}/m)\Delta t + (\dot{\boldsymbol{p}}_{j+1}^{\text{fwd}}/2m)\Delta t^2 + \mathbf{rem}_j^{\text{fwd},3}) \right\|_2 \\
&\le \left\| R(\boldsymbol{q}_{j+1}^{\text{rev}}) - \boldsymbol{q}_{j+1}^{\text{fwd}} \right\|_2 + \frac{1}{m}\left\| R(\boldsymbol{p}_{j+1}^{\text{rev}}) - \boldsymbol{p}_{j+1}^{\text{fwd}} \right\|_2\Delta t + \frac{1}{2m}\left\| R(\dot{\boldsymbol{p}}_{j+1}^{\text{rev}}) - \dot{\boldsymbol{p}}_{j+1}^{\text{fwd}} \right\|_2\Delta t^2 + \|\mathbf{rem}_j^{\text{rev},3}\|_2 + \|\mathbf{rem}_j^{\text{fwd},3}\|_2 \\
&\le \left[ C_3^{\text{rev}}(T-j-1)^3 + \frac{C_2^{\text{rev}}}{m}(T-j-1)^2 + \frac{C_1^{\text{rev}}}{2m}(T-j-1) + 2c_3 \right]\Delta t^3,
\end{aligned}
$$

where the first inequality uses the triangle inequality, and in the second inequality we use (24a), (24b) and (24c). Thus taking $C_3^{\text{rev}} = \max\{C_2^{\text{rev}}/3m, C_1^{\text{rev}}/6m, 2c_3\}$, we have the final expression above is upper bounded by $C_3^{\text{rev}}(T-j)^3\Delta t^3$, and so the claim holds for $j$.

Since both the base case and the inductive step have been proven, by the principle of mathematical induction, (23b), (23a) and (23c) hold for all $k = T, T-1, \cdots, 0$.

With this we finish the proof by plugging (23b) and (23a) into the loss function:

$$
\begin{aligned}
\sum_{j=0}^T \left\| R(\boldsymbol{z}_j^{\text{rev}}) - \boldsymbol{z}_j^{\text{fwd}} \right\|_2^2 &= \sum_{j=0}^T \left\| R(\boldsymbol{p}_j^{\text{rev}}) - \boldsymbol{p}_j^{\text{fwd}} \right\|_2^2 + \sum_{j=0}^T \left\| R(\boldsymbol{q}_j^{\text{rev}}) - \boldsymbol{q}_j^{\text{fwd}} \right\|_2^2 \\
&\le \left( C_2^{\text{rev}} \right)^2 \sum_{j=0}^T (T-j)^4 \Delta t^4 + \left( C_3^{\text{rev}} \right)^2 \sum_{j=0}^T (T-j)^6 \Delta t^6 \\
&= \mathcal{O}(T^5 \Delta t^4).
\end{aligned}
\tag{25}
$$

## A.4 Proof of Lemma 3.2

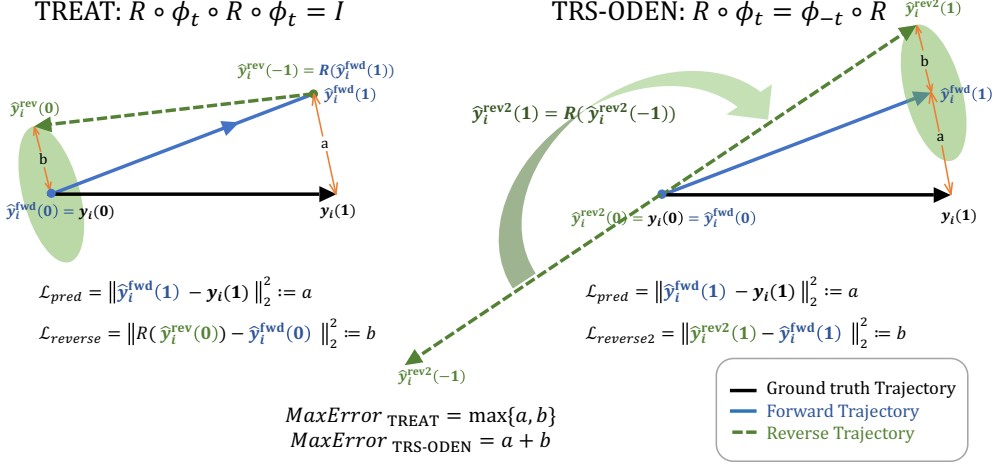

Figure 8: Comparison between two reversal loss implementation

We expect an ideal model to align both the predicted forward and reverse trajectories with the ground truth. As shown in Figure 8, we integrate one step from the initial state $\hat{\boldsymbol{y}}_i^{\text{fwd}}(0)$ (which is the same as $\boldsymbol{y}_i(0)$) and reach the state $\hat{\boldsymbol{y}}_i^{\text{fwd}}(1)$.

The first reverse loss implementation (ours) follows Lemma 2.1 as $R \circ \Phi_t \circ R \circ \Phi_t = I$, which means when we evolve forward and reach the state $\hat{\boldsymbol{y}}_i^{\text{fwd}}(1)$ we reverse it into $\hat{\boldsymbol{y}}_i^{\text{rev}}(-1) = R(\hat{\boldsymbol{y}}_i^{\text{fwd}}(1))$ and go back to reach $\hat{\boldsymbol{y}}_i^{\text{rev}}(0)$, then reverse it to get $R(\hat{\boldsymbol{y}}_i^{\text{rev}}(0))$, which ideally should be the same as $\hat{\boldsymbol{y}}_i^{\text{fwd}}(0)$.

The second reverse loss implementation follows Eqn 5 as $R \circ \Phi_t = \Phi_{-t} \circ R$, which means we first reverse the initial state as $\hat{\boldsymbol{y}}_i^{\text{rev2}}(0) = R(\boldsymbol{y}_i(0))$, then evolve the reverse trajectory in the opposite direction to reach $\hat{\boldsymbol{y}}_i^{\text{rev2}}(-1)$, and then perform a symmetric operation to reach $\hat{\boldsymbol{y}}_i^{\text{rev2}}(1)$, aligning it with the forward trajectory.

We assume the two reconstruction losses $\mathcal{L}_{pred} = \|\hat{\boldsymbol{y}}_i^{\text{fwd}}(1) - \boldsymbol{y}_i(1)\|_2^2 := a$ are the same. For the time-reversal losses, we also assume they have reached the same value $b$:

$$\mathcal{L}_{reverse} = \|R(\hat{\boldsymbol{y}}_i^{\text{rev}}(0)) - \hat{\boldsymbol{y}}_i^{\text{fwd}}(0)\|_2^2 + \|R(\hat{\boldsymbol{y}}_i^{\text{rev}}(-1)) - \hat{\boldsymbol{y}}_i^{\text{fwd}}(1)\|_2^2 = \|R(\hat{\boldsymbol{y}}_i^{\text{rev}}(0)) - \hat{\boldsymbol{y}}_i^{\text{fwd}}(0)\|_2^2 := b,$$

$$\mathcal{L}_{reverse2} = \|\hat{\boldsymbol{y}}_i^{\text{rev2}}(0) - \hat{\boldsymbol{y}}_i^{\text{fwd}}(0)\|_2^2 + \|\hat{\boldsymbol{y}}_i^{\text{rev2}}(1) - \hat{\boldsymbol{y}}_i^{\text{fwd}}(1)\|_2^2 = \|\hat{\boldsymbol{y}}_i^{\text{rev2}}(1) - \hat{\boldsymbol{y}}_i^{\text{fwd}}(1)\|_2^2 := b,$$

As shown in Figure 8 where we illustrate the worst case scenario $MaxError_{gt\_rev} = \max_{k \in [K]} \|\boldsymbol{y}_i(t_k) - \hat{\boldsymbol{y}}_i^{\text{rev}}(t'_{K-k})\|_2$ of TREAT and TRS-ODEN, we can see that in our implementation the worst error is the maximum of two loss, while the TRS-ODEN's implementation has the risk of accumulating the error together, making the worst error being the sum of both:

$$MaxError_{\text{TREAT}} = \max\left\{\left\|R(\hat{\boldsymbol{y}}_i^{\text{rev}}(0)) - \boldsymbol{y}_i(0)\right\|_2, \left\|R(\hat{\boldsymbol{y}}_i^{\text{rev}}(-1)) - \boldsymbol{y}_i(1)\right\|_2\right\} = max\{a, b\},$$

$$MaxError_{\text{TRS-ODEN}} = \max\left\{\left\|\hat{\boldsymbol{y}}_i^{\text{rev2}}(0) - \boldsymbol{y}_i(0)\right\|_2, \left\|\hat{\boldsymbol{y}}_i^{\text{rev2}}(1) - \boldsymbol{y}_i(1)\right\|_2\right\}$$
$$= \max\left\{0, \left\|R(\hat{\boldsymbol{y}}_i^{\text{rev}}(-1)) - \boldsymbol{y}_i(1)\right\|_2\right\}$$
$$= \left\|\hat{\boldsymbol{y}}_i^{\text{rev2}}(1) - \hat{\boldsymbol{y}}_i^{\text{fwd}}(1)\right\|_2 + \left\|\hat{\boldsymbol{y}}^{\text{fwd}}(1) - \boldsymbol{y}(1)\right\|_2 = a + b,$$

$$(26)$$

So it is obvious that $MaxError_{\text{TREAT}}$ made by TREAT is smaller., which means our model achieves a smaller error of the maximum distance between the reversal and ground truth trajectory.

## B  Example of varying dynamical systems

We illustrate the energy conservation and time reversal of the three n-body spring systems used in our experiments. We use the Hamiltonian formalism of systems under classical mechanics to describe their dynamics and verify their energy conservation and time-reversibility characteristics.

The scalar function that describes a system's motion is called the Hamiltonian, $\mathcal{H}$, and is typically equal to the total energy of the system, that is, the potential energy plus the kinetic energy (North, 2021). It describes the phase space equations of motion by following two first-order ODEs called Hamilton's equations:

$$\frac{d\mathbf{q}}{dt} = \frac{\partial \mathcal{H}(\mathbf{q}, \mathbf{p})}{\partial \mathbf{p}}, \frac{d\mathbf{p}}{dt} = -\frac{\partial \mathcal{H}(\mathbf{q}, \mathbf{p})}{\partial \mathbf{q}}, \tag{27}$$

where $\mathbf{q} \in \mathbb{R}^n, \mathbf{p} \in \mathbb{R}^n$, and $\mathcal{H} : \mathbb{R}^{2n} \mapsto \mathbb{R}$ are positions, momenta, and Hamiltonian of the system.

Under this formalism, energy conservative is defined by $d\mathcal{H}/dt = 0$, and the time-reversal symmetry is defined by $\mathcal{H}(q, p, t) = \mathcal{H}(q, -p, -t)$ (Lamb and Roberts, 1998).

### B.1  Conservative and reversible systems.

A simple example is the isolated n-body spring system, which can be described by :

$$\frac{d\mathbf{q_i}}{dt} = \frac{\mathbf{p_i}}{m}$$
$$\frac{d\mathbf{p_i}}{dt} = \sum_{j \in N_i} -k(\mathbf{q_i} - \mathbf{q_j}), \tag{28}$$

where $\mathbf{q} = (\mathbf{q_1}, \mathbf{q_2}, \cdots, \mathbf{q_N})$ is a set of positions of each object , $\mathbf{p} = (\mathbf{p_1}, \mathbf{p_2}, \cdots, \mathbf{p_N})$ is a set of momenta of each object, $m_i$ is mass of each object, $k$ is spring constant.

The Hamilton's equations are:

$$\frac{\partial \mathcal{H}(\mathbf{q}, \mathbf{p})}{\partial \mathbf{p_i}} = \frac{d\mathbf{q_i}}{dt} = \frac{\mathbf{p_i}}{m}$$

$$\frac{\partial \mathcal{H}(\mathbf{q}, \mathbf{p})}{\partial \mathbf{q_i}} = -\frac{d\mathbf{p_i}}{dt} = \sum_{j \in N_i} k(\mathbf{q_i} - \mathbf{q_j}), \tag{29}$$

Hence, we can obtain the Hamiltonian through the integration of the above equation.

$$\mathcal{H}(\mathbf{q}, \mathbf{p}) = \sum_{i=1}^{N} \frac{\mathbf{p_i}^2}{2m_i} + \frac{1}{2} \sum_{i=1}^{N} \sum_{j \in N_i}^{N} \frac{1}{2} k(\mathbf{q_i} - \mathbf{q_j})^2, \tag{30}$$

**Verify the systems' energy conservation**

$$\frac{d\mathcal{H}(\mathbf{q}, \mathbf{p})}{dt} = \frac{1}{dt} \left( \sum_{i=1}^{N} \frac{\mathbf{p_i}^2}{2m_i} \right) + \frac{1}{dt} \left( \frac{1}{2} \sum_{i=1}^{N} \sum_{j \in N_i}^{N} \frac{1}{2} k(\mathbf{q_i} - \mathbf{q_j})^2 \right) = 0, \tag{31}$$

So it is conservative.

**Verify the systems' time-reversal symmetry** We do the transformation $R : (\mathbf{q}, \mathbf{p}, t) \mapsto (\mathbf{q}, -\mathbf{p}, -t)$.

$$\mathcal{H}(\mathbf{q}, \mathbf{p}) = \sum_{i=1}^{N} \frac{\mathbf{p_i}^2}{2m_i} + \frac{1}{2} \sum_{i=1}^{N} \sum_{j \in N_i}^{N} \frac{1}{2} k(\mathbf{q_i} - \mathbf{q_j})^2,$$

$$\mathcal{H}(\mathbf{q}, -\mathbf{p}) = \sum_{i=1}^{N} \frac{(-\mathbf{p_i})^2}{2m_i} + \frac{1}{2} \sum_{i=1}^{N} \sum_{j \in N_i}^{N} \frac{1}{2} k(\mathbf{q_i} - \mathbf{q_j})^2, \tag{32}$$

It is obvious $\mathcal{H}(\mathbf{q}, \mathbf{p}) = \mathcal{H}(\mathbf{q}, -\mathbf{p})$, so it is reversible

## B.2 Non-conservative and reversible systems.

A simple example is a n-body spring system with periodical external force, which can be described by:

$$\frac{d\mathbf{q_i}}{dt} = \frac{\mathbf{p_i}}{m}$$

$$\frac{d\mathbf{p_i}}{dt} = \sum_{j \in N_i}^{N} -k(\mathbf{q_i} - \mathbf{q_j}) - k_1 \cos \omega t, \tag{33}$$

The Hamilton's equations are:

$$\frac{\partial \mathcal{H}(\mathbf{q}, \mathbf{p})}{\partial \mathbf{p_i}} = \frac{d\mathbf{q_i}}{dt} = \frac{\mathbf{p_i}}{m}$$

$$\frac{\partial \mathcal{H}(\mathbf{q}, \mathbf{p})}{\partial \mathbf{q_i}} = -\frac{d\mathbf{p_i}}{dt} = \sum_{j \in N_i} k(\mathbf{q_i} - \mathbf{q_j}) + k_1 \cos \omega t, \tag{34}$$

Hence, we can obtain the Hamiltonian through the integration of the above equation:

$$\mathcal{H}(\mathbf{q}, \mathbf{p}) = \sum_{i=1}^{N} \frac{\mathbf{p_i}^2}{2m_i} + \frac{1}{2} \sum_{i=1}^{N} \sum_{j \in N_i}^{N} \frac{1}{2} k(\mathbf{q_i} - \mathbf{q_j})^2 + \sum_{i=1}^{N} q_i * k_1 \cos \omega t, \tag{35}$$

**Verify the systems' energy conservation**

$$\frac{d\mathcal{H}(\mathbf{q}, \mathbf{p})}{dt} = \frac{1}{dt} \left( \sum_{i=1}^{N} \frac{\mathbf{p_i}^2}{2m_i} \right) + \frac{1}{dt} \left( \frac{1}{2} \sum_{i=1}^{N} \sum_{j \in N_i}^{N} \frac{1}{2} k(\mathbf{q_i} - \mathbf{q_j})^2 \right) + \frac{1}{dt} \left( \sum_{i=1}^{N} q_i * k_1 \cos \omega t \right)$$

$$= 0 + \frac{1}{dt} \left( \sum_{i=1}^{N} q_i k_1 \cos \omega t \right) \tag{36}$$

$$= \left( \sum_{i=1}^{N} -\omega q_i k_1 \sin \omega t \right) \neq 0$$

So it is non-conservative.

**Verify the systems' time-reversal symmetry** We do the transformation $R : (\mathbf{q}, \mathbf{p}, t) \mapsto (\mathbf{q}, -\mathbf{p}, -t)$.

$$
\mathcal{H}(\mathbf{q},\mathbf{p}) = \sum_{i=1}^{N} \frac{\mathbf{p_i}^2}{2m_i} + \frac{1}{2}\sum_{i=1}^{N}\sum_{j\in N_i}^{N} \frac{1}{2}k(\mathbf{q_i} - \mathbf{q_j})^2 + \sum_{i=1}^{N} q_i * k_1 \cos\omega t,
$$

$$
\mathcal{H}(\mathbf{q},-\mathbf{p}) = \sum_{i=1}^{N} \frac{(-\mathbf{p_i})^2}{2m_i} + \frac{1}{2}\sum_{i=1}^{N}\sum_{j\in N_i}^{N} \frac{1}{2}k(\mathbf{q_i} - \mathbf{q_j})^2 + \sum_{i=1}^{N} q_i * k_1 \cos\omega(-t),
$$

(37)

It is obvious $\mathcal{H}(\mathbf{q}, \mathbf{p}, t) = \mathcal{H}(\mathbf{q}, -\mathbf{p}, t)$, so it is reversible

### B.3 Non-conservative and irreversible systems.

A simple example is an n-body spring system with frictions proportional to its velocity,$\gamma$ is the coefficient of friction, which can be described by:

$$
\begin{aligned}
\frac{d\mathbf{q}_i}{dt} &= \frac{\mathbf{p}_i}{m} \\
\frac{d\mathbf{p}_i}{dt} &= -k_0\mathbf{q}_i - \gamma\frac{\mathbf{p}_i}{m}
\end{aligned}
$$

(38)

The Hamilton's equations are:

$$
\begin{aligned}
\frac{\partial \mathcal{H}(\mathbf{q}, \mathbf{p})}{\partial \mathbf{p_i}} &= \frac{d\mathbf{q_i}}{dt} = \frac{\mathbf{p_i}}{m} \\
\frac{\partial \mathcal{H}(\mathbf{q}, \mathbf{p})}{\partial \mathbf{q_i}} &= -\frac{d\mathbf{p_i}}{dt} = \sum_{j\in N_i} k(\mathbf{q_i} - \mathbf{q_j}) + \gamma\frac{\mathbf{p_i}}{m}
\end{aligned}
$$

(39)

Hence, we can obtain the Hamiltonian through the integration of the above equation:

$$
\mathcal{H}(\mathbf{q}, \mathbf{p}) = \sum_{i=1}^{N} \frac{\mathbf{p_i}^2}{2m_i} + \frac{1}{2}\sum_{i=1}^{N}\sum_{j\in N_i}^{N} \frac{1}{2}k(\mathbf{q_i} - \mathbf{q_j})^2 + \sum_{i=1}^{N} \frac{\gamma}{m}\int_0^t \frac{\mathbf{p_i}^2}{m}dt,
$$

(40)

**Verify the systems' energy conservation**

$$
\begin{aligned}
\frac{d\mathcal{H}(\mathbf{q}, \mathbf{p})}{dt} &= \frac{1}{dt}(\sum_{i=1}^{N} \frac{\mathbf{p_i}^2}{2m_i}) + \frac{1}{dt}(\frac{1}{2}\sum_{i=1}^{N}\sum_{j\in N_i}^{N} \frac{1}{2}k(\mathbf{q_i} - \mathbf{q_j})^2) + \frac{1}{dt}(\sum_{i=1}^{N} \frac{\gamma}{m}\int_0^t \frac{\mathbf{p_i}^2}{m}dt) \\
&= 0 + \frac{1}{dt}(\sum_{i=1}^{N} \frac{\gamma}{m}\int_0^t \frac{\mathbf{p_i}^2}{m}dt) \\
&= (\sum_{i=1}^{N} \frac{\gamma}{m}\frac{\mathbf{p_i}^2}{m}) \neq 0
\end{aligned}
$$

(41)

So it is non-conservative.

**Verify the systems' time-reversal symmetry** We do the transformation $R : (\mathbf{q}, \mathbf{p}, t) \mapsto (\mathbf{q}, -\mathbf{p}, -t)$.

$$
\mathcal{H}(\mathbf{q}, \mathbf{p}) = \sum_{i=1}^{N} \frac{\mathbf{p_i}^2}{2m_i} + \frac{1}{2}\sum_{i=1}^{N}\sum_{j\in N_i}^{N} \frac{1}{2}k(\mathbf{q_i} - \mathbf{q_j})^2 + \sum_{i=1}^{N} \frac{\gamma}{m}\int_0^t \frac{\mathbf{p_i}^2}{m}dt,
$$

$$
\mathcal{H}(\mathbf{q}, -\mathbf{p}) = \sum_{i=1}^{N} \frac{(-\mathbf{p_i})^2}{2m_i} + \frac{1}{2}\sum_{i=1}^{N}\sum_{j\in N_i}^{N} \frac{1}{2}k(\mathbf{q_i} - \mathbf{q_j})^2 + \sum_{i=1}^{N} \frac{\gamma}{m}\int_0^{(-t)} \frac{\mathbf{p_i}^2}{m}d(-t),
$$

(42)

It is obvious $\mathcal{H}(\mathbf{q}, \mathbf{p}, t) \neq \mathcal{H}(\mathbf{q}, -\mathbf{p}, t)$, so it is irreversible

# C  Dataset

In our experiments, all datasets are synthesized from ground-truth physical law via sumulation. We generate five simulated datasets: three $n$-body spring systems under damping, periodic, or no external force, one chaotic tripe pendulum dataset with three sequentially connected stiff sticks that form and a chaotic strange attractor. We name the first three as *Sipmle Spring*, *Forced Spring*, and *Damped Spring* respectively. For multi-agent systems, all $n$-body spring systems contain 5 interacting balls, with varying connectivities. Each *Pendulum* system contains 3 connected stiff sticks. For single-agent systems, all spring systems contain only one ball. For the chaotic single *Attractor*, we follow the setting of  (Huh et al., 2020).

For the $n$-body spring system, we randomly sample whether a pair of objects are connected, and model their interaction via forces defined by Hooke's law. In the *Damped spring*, the objects have an additional friction force that is opposite to their moving direction and whose magnitude is proportional to their speed. In the *Forced spring*, all objects have the same external force that changes direction periodically. We show in Figure 1(a), the energy variation in both of the *Damped spring* and *Forced spring* is significant. For the chaotic triple *Pendulum* , the equations governing the motion are inherently nonlinear. Although this system is deterministic, it is also highly sensitive to the initial condition and numerical errors (Shinbrot et al., 1992; Awrejcewicz et al., 2008; Stachowiak and Okada, 2006). This property is often referred to as the "butterfly effect", as depicted in Figure 9. Unlike for $n$-body spring systems, where the forces and equations of motion can be easily articulated, for the *Pendulum*, the explicit forces cannot be directly defined, and the motion of objects can only be described through Lagrangian formulations (North, 2021), making the modeling highly complex and raising challenges for accurate learning. We simulate the trajectories by using Euler's method for

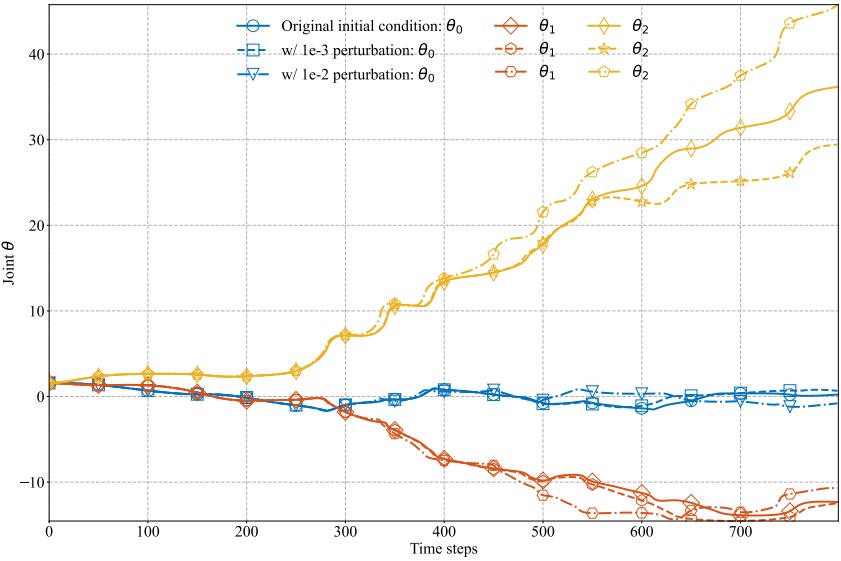

Figure 9: Illustration to show the pendulum is highly-sensitive to initial states

$n$-body spring systems and using the 4th order Runge-Kutta (RK4) method for the *Pendulum* and *Attractor* . For all spring systems and *Pendulum*, We integrate with a fixed step size and subsample every 100 steps. For training, we use a total of 6000 forward steps. To generate irregularly sampled partial observations, we follow (Huang et al., 2020) and sample the number of observations n from a uniform distribution U(40, 52) and draw the n observations uniformly for each object. For testing, we additionally sample 40 observations following the same procedure from PDE steps [6000, 12000], besides generating observations from steps [1, 6000]. The above sampling procedure is conducted independently for each object. We generate 20k training samples and 5k testing samples for each dataset. For *Attractor*, we integrate a total of 600 forward steps for training and subsample every 10 steps. For testing, we additionally sample 40 observations from step [600,1200].The irregularly sampled partial observations generation is the same as above. We generate 1000 training samples and 50 testing samples following  (Huh et al., 2020). Therefore, for all datasets, condition length is

60 steps and prediction length is 40s steps. The features (position/velocity) are normalized to the maximum absolute value of 1 across training and testing datasets.

We also compute the Maximum Lyapunov Exponent (MLE) to assess the chaos level of the systems, using the formula:

$$\lambda = max_{t\to\inf}(\frac{1}{t}\ln\frac{||\delta(t)||}{||\delta(0)||}).$$

We set fixed initial values for each dataset and generate 10 trajectories by perturbing the initial values with random noise (0, 0.0001). We calculate the Maximum Lyapunov Exponent (MLE) between any two trajectories. Finally, we compute the average and std of MLE from all pairs to gauge the chaotic behavior of each dataset. The data is presented in the table below:

Table 2: MLE of different Multi-agent Systems

| Dataset | Simple Spring | Forced Spring | Damped Spring | Pendulum |
|---|---|---|---|---|
| MLE(in 60 steps) | $0.4031 \pm 0.3944$ | $1.0087 \pm 1.0577$ | $0.6307 \pm 0.7065$ | $34.1832 \pm 30.1846$ |

From the table, it's evident that the order of MLE values is: *Pendulum* » three *Spring* datasets. This observation is consistent with the evaluation results based on MSE presented in our previous responses in Table 3 which indicates that as the prediction length(steps*step size) increases, there is a more significant performance degradation of all models on *Pendulum* dataset.

In the following subsections, we show the dynamical equations of each dataset in detail.

## C.1 Spring Systems

### C.1.1 Simple Spring

The dynamical equations of *simple spring* are as follows:

$$\frac{d\mathbf{q_i}}{dt} = \frac{\mathbf{p_i}}{m}$$
$$\frac{d\mathbf{p_i}}{dt} = \sum_{j\in N_i}^{N} -k(\mathbf{q_i} - \mathbf{q_j}) \tag{43}$$

where where $\mathbf{q} = (\mathbf{q_1}, \mathbf{q_2}, \cdots, \mathbf{q_N})$ is a set of positions of each object , $\mathbf{p} = (\mathbf{p_1}, \mathbf{p_2}, \cdots, \mathbf{p_N})$ is a set of momenta of each object. We set the mass of each object $m = 1$, the spring constant $k = 0.1$.

### C.1.2 Damped Spring

The dynamical equations of *damped spring* are as follows:

$$\frac{d\mathbf{q_i}}{dt} = \frac{\mathbf{p_i}}{m}$$
$$\frac{d\mathbf{p_i}}{dt} = \sum_{j\in N_i} -k(\mathbf{q_i} - \mathbf{q_j}) - \gamma\frac{\mathbf{p}_i}{m} \tag{44}$$

where where $\mathbf{q} = (\mathbf{q_1}, \mathbf{q_2}, \cdots, \mathbf{q_N})$ is a set of positions of each object, $\mathbf{p} = (\mathbf{p_1}, \mathbf{p_2}, \cdots, \mathbf{p_N})$ is a set of momenta of each object, We set the mass of each object $m = 1$, the spring constant $k = 0.1$, the coefficient of friction $\gamma = 10$.

### C.1.3 Forced Spring

The dynamical equations of *forced spring* system are as follows:

$$\frac{d\mathbf{q_i}}{dt} = \frac{\mathbf{p_i}}{m}$$
$$\frac{d\mathbf{p_i}}{dt} = \sum_{j\in N_i}^{N} -k(\mathbf{q_i} - \mathbf{q_j}) - k_1\cos\omega t, \tag{45}$$

where where $\mathbf{q} = (\mathbf{q_1}, \mathbf{q_2}, \cdots, \mathbf{q_N})$ is a set of positions of each object , $\mathbf{p} = (\mathbf{p_1}, \mathbf{p_2}, \cdots, \mathbf{p_N})$ is a set of momenta of each object. We set the mass of each object $m = 1$ , the spring constant $k = 0.1$, the external strength $k_1 = 10$ and the frequency of variation $\omega = 1$

We simulate the positions and momentums of three spring systems by using Euler methods as follows:

$$\mathbf{q_i}(t+1) = \mathbf{q_i}(t) + \frac{d\mathbf{q_i}}{dt}\Delta t$$

$$\mathbf{p_i}(t+1) = \mathbf{p_i}(t) + \frac{d\mathbf{p_i}}{dt}\Delta t$$

(46)

where $\frac{d\mathbf{q_i}}{dt}$ and $\frac{d\mathbf{p_i}}{dt}$ were defined as above for each datasets, and $\Delta t = 0.001$ is the integration steps.

## C.2 Chaotic Pendulum

In this section, we demonstrate how to derive the dynamics equations for a chaotic triple pendulum using the Lagrangian formalism.

The moment of inertia of each stick about the centroid is

$$I = \frac{1}{12}ml^2$$

(47)

The position of the center of gravity of each stick is as follows:

$$x_1 = \frac{l}{2}\sin\theta_1, \quad y_1 = -\frac{l}{2}\cos\theta_1$$

$$x_2 = l(\sin\theta_1 + \frac{1}{2}\sin\theta_2), \quad y_2 = -l(\cos\theta_1 + \frac{1}{2}\cos\theta_2)$$

$$x_3 = l(\sin\theta_1 + \sin\theta_2 + \frac{1}{2}\sin\theta_3), \quad y_3 = -l(\cos\theta_1 + \cos\theta_2 + \frac{1}{2}\cos\theta_3)$$

(48)

The change in the center of gravity of each stick is:

$$\dot{x}_1 = \frac{l}{2}\cos\theta_1 \cdot \dot{\theta}_1, \quad \dot{y}_1 = \frac{l}{2}\sin\theta_1 \cdot \dot{\theta}_1$$

$$\dot{x}_2 = l(\cos\theta_1 \cdot \dot{\theta}_1 + \frac{1}{2}\cos\theta_2 \cdot \dot{\theta}_2), \quad \dot{y}_2 = l(\sin\theta_1 \cdot \dot{\theta}_1 + \frac{1}{2}\sin\theta_2 \cdot \dot{\theta}_2)$$

(49)

$$\dot{x}_3 = l(\cos\theta_1 \cdot \dot{\theta}_1 + \cos\theta_2 \cdot \dot{\theta}_2 + \frac{1}{2}\cos\theta_3 \cdot \dot{\theta}_3), \quad \dot{y}_3 = l(\sin\theta_1 \cdot \dot{\theta}_1 + \sin\theta_2 \cdot \dot{\theta}_2 + \frac{1}{2}\sin\theta_3 \cdot \dot{\theta}_3)$$

The Lagrangian L of this triple pendulum system is:

$$\begin{aligned}
\mathcal{L} =& T - V \\
=& \frac{1}{2}m(\dot{x_1}^2 + \dot{x_2}^2 + \dot{x_3}^2 + \dot{y_1}^2 + \dot{y_2}^2 + \dot{y_3}^2) + \frac{1}{2}I(\dot{\theta_1}^2 + \dot{\theta_2}^2 + \dot{\theta_3}^2) - mg(y_1 + y_2 + y_3) \\
=& \frac{1}{6}ml(9\dot{\theta}_2\dot{\theta}_1 l \cos(\theta_1 - \theta_2) + 3\dot{\theta}_3\dot{\theta}_1 l \cos(\theta_1 - \theta_3) + 3\dot{\theta}_2\dot{\theta}_3 l \cos(\theta_2 - \theta_3) + 7\dot{\theta}_1^2 l + 4\dot{\theta}_2^2 l + \dot{\theta}_3^2 l \\
& + 15g\cos(\theta_1) + 9g\cos(\theta_2) + 3g\cos(\theta_3))
\end{aligned}$$

(50)

The Lagrangian equation is defined as follows:

$$\frac{d}{dt}\frac{\partial\mathcal{L}}{\partial\dot{\theta}} - \frac{\partial\mathcal{L}}{\partial\theta} = \mathbf{0}$$

(51)

and we also have:

$$\frac{\partial\mathcal{L}}{\partial\dot{\theta}} = \frac{\partial T}{\partial\dot{\theta}} = p$$

$$\dot{p} = \frac{d}{dt}\frac{\partial\mathcal{L}}{\partial\dot{\theta}} = \frac{\partial\mathcal{L}}{\partial\theta}$$

(52)

where p is the Angular Momentum.
We can list the equations for each of the three sticks separately:

$$p_1 = \frac{\partial\mathcal{L}}{\partial\dot{\theta}_\mathbf{1}} \quad \dot{p}_1 = \frac{\partial\mathcal{L}}{\partial\theta_\mathbf{1}}$$

$$p_2 = \frac{\partial\mathcal{L}}{\partial\dot{\theta}_\mathbf{2}} \quad \dot{p}_2 = \frac{\partial\mathcal{L}}{\partial\theta_\mathbf{2}}$$

$$p_3 = \frac{\partial\mathcal{L}}{\partial\dot{\theta}_\mathbf{3}} \quad \dot{p}_3 = \frac{\partial\mathcal{L}}{\partial\theta_\mathbf{3}}$$

(53)

Finally, we have :

$$
\begin{cases}
\dot{\theta}_1 = \frac{6(9p_1\cos(2(\theta_2-\theta_3))+27p_2\cos(\theta_1-\theta_2)-9p_2\cos(\theta_1+\theta_2-2\theta_3)+21p_3\cos(\theta_1-\theta_3)-27p_3\cos(\theta_1-2\theta_2+\theta_3)-23p_1)}{ml^2(81\cos(2(\theta_1-\theta_2))-9\cos(2(\theta_1-\theta_3))+45\cos(2(\theta_2-\theta_3))-169)} \\
\dot{\theta}_2 = \frac{6(27p_1\cos(\theta_1-\theta_2)-9p_1\cos(\theta_1+\theta_2-2\theta_3)+9p_2\cos(2(\theta_1-\theta_3))-27p_3\cos(2\theta_1-\theta_2-\theta_3)+57p_3\cos(\theta_2-\theta_3)-47p_2)}{ml^2(81\cos(2(\theta_1-\theta_2))-9\cos(2(\theta_1-\theta_3))+45\cos(2(\theta_2-\theta_3))-169)} \\
\dot{\theta}_3 = \frac{6(21p_1\cos(\theta_1-\theta_3)-27p_1\cos(\theta_1-2\theta_2+\theta_3)-27p_2\cos(2\theta_1-\theta_2-\theta_3)+57p_2\cos(\theta_2-\theta_3)+81p_3\cos(2(\theta_1-\theta_2))-143p_3)}{ml^2(81\cos(2(\theta_1-\theta_2))-9\cos(2(\theta_1-\theta_3))+45\cos(2(\theta_2-\theta_3))-169)} \\
\dot{p}_1 = -\frac{1}{2}ml\left(3\dot{\theta}_2\dot{\theta}_1 l\sin(\theta_1-\theta_2)+\dot{\theta}_1\dot{\theta}_3 l\sin(\theta_1-\theta_3)+5g\sin(\theta_1)\right) \\
\dot{p}_1 = -\frac{1}{2}ml\left(-3\dot{\theta}_1\dot{\theta}_2 l\sin(\theta_1-\theta_2)+\dot{\theta}_2\dot{\theta}_3 l\sin(\theta_2-\theta_3)+3g\sin(\theta_2)\right) \\
\dot{p}_1 = -\frac{1}{2}ml\left(\dot{\theta}_1\dot{\theta}_3 l\sin(\theta_1-\theta_3)+\dot{\theta}_2\dot{\theta}_3 l\sin(\theta_2-\theta_3)-g\sin(\theta_3)\right)
\end{cases}
\tag{54}
$$

We simulate the angular of the three sticks by using the Runge-Kutta 4th Order Method as follows:

$$
\begin{aligned}
\Delta\boldsymbol{\theta}^1(t) &= \dot{\boldsymbol{\theta}}(t,\boldsymbol{\theta}(t))\cdot\Delta t \\
\Delta\boldsymbol{\theta}^2(t) &= \dot{\boldsymbol{\theta}}(t+\frac{\Delta t}{2},\boldsymbol{\theta}(t)+\frac{\Delta\boldsymbol{\theta}^1(t)}{2})\cdot\Delta t \\
\Delta\boldsymbol{\theta}^3(t) &= \dot{\boldsymbol{\theta}}(t+\frac{\Delta t}{2},\boldsymbol{\theta}(t)+\frac{\Delta\boldsymbol{\theta}^2(t)}{2})\cdot\Delta t \\
\Delta\boldsymbol{\theta}^4(t) &= \dot{\boldsymbol{\theta}}(t+\Delta t,\boldsymbol{\theta}(t)+\Delta\boldsymbol{\theta}^3(t))\cdot\Delta t \\
\Delta\boldsymbol{\theta}(t) &= \frac{1}{6}(\Delta\boldsymbol{\theta}^1(t)+\Delta\boldsymbol{\theta}^2(t)+\Delta\boldsymbol{\theta}^3(t)+\Delta\boldsymbol{\theta}^4(t)) \\
\boldsymbol{\theta}(t+1) &= \boldsymbol{\theta}(t)+\Delta\boldsymbol{\theta}(t)
\end{aligned}
\tag{55}
$$

where $\dot{\boldsymbol{\theta}}$ was defined as above , and $\Delta t = 0.0001$ is the integration steps.

## C.3 Chaotic Strange Attractor

The dynamical equations of this reversible strange attractor are as follows:

$$
\begin{aligned}
\frac{dx}{dt} &= 1+yz, \\
\frac{dy}{dt} &= -xz, \\
\frac{dz}{dt} &= y^2+2yz, \\
x,y,x &\in \mathbb{R}
\end{aligned}
\tag{56}
$$

The above equations can be presented as $(\dot{x}(t),\dot{y}(t),\dot{z}(t)) = Dynamic(x(t),y(t),z(t))$.
We simulate $K(t) = (x(t),y(t),z(t))$ by using the Runge-Kutta 4th Order Method as follows:

$$
\begin{aligned}
\Delta K_1(t) &= Dynamic(K(t))*\Delta t \\
\Delta K_2(t) &= Dynamic(K(t)+\frac{\Delta K_1(t)}{2})*\Delta t \\
\Delta K_3(t) &= Dynamic(K(t)+\frac{\Delta K_2(t)}{2})*\Delta t \\
\Delta K_4(t) &= Dynamic(K(t)+\Delta K_3(t))*\Delta t \\
\Delta K(t) &= \frac{1}{6}(\Delta K_1(t)+\Delta K_2(t)+\Delta K_3(t)+\Delta K_4(t)) \\
K(t+1) &= K(t)+\Delta K(t)
\end{aligned}
\tag{57}
$$

We sampling $z(t_0)$ randomly from uniform distribution [1, 3] while fixing $x(t_0) = y(t_0) = 0$. We set the trajectory lengths of both training and test dataset to 600, with regular time-step size $\Delta t = 0.03$ and the sample frequency of 10. We add Gaussian noise $0.05n$, $n \sim \mathcal{N}(0,1)$ to training trajectories.

## C.4 Human Motion

For the real-world motion capture dataset(Carnegie Mellon University, 2003), we focus on the walking sequences of subject 35. Each sample in this dataset is represented by 31 trajectories, each corresponding to the movement of a single joint. For each joint, we first randomly sample the number of observations from a uniform distribution $\mathcal{U}(30, 42)$ and then sample uniformly from the first 50 frames for training and validation trajectories. For testing, we additionally sampled 40 observations from frames $[51, 99]$.We split different walking sequences into training (15 trials) and test sets (7 trials). For each walking sequence, we further split it into several non-overlapping small sequences with maximum length 50 for training, and maximum length 100 for testing. In this way, we generate total 120 training samples and 27 testing samples. We normalize all features (position/velocity) to maximum absolute value of 1 across training and testing datasets.

# D  Model Details

In the following we introduce in details how we implement our model and each baseline.

## D.1  Initial State Encoder

For multi-agent systems, the initial state encoder computes the latent node initial states $z_i(t)$ for all agents simultaneously considering their mutual interaction. Specifically, it first fuses all observations into a temporal graph and conducts dynamic node representation through a spatial-temporal GNN as in (Huang et al., 2020):

$$
\begin{aligned}
\boldsymbol{h}_{j(t)}^{l+1} &= \boldsymbol{h}_{j(t)}^{l} + \sigma\left(\sum_{i(t')\in\mathcal{N}_{j(t)}} \alpha_{i(t')\to j(t)}^{l} \times \boldsymbol{W}_v \hat{\boldsymbol{h}}_{i(t')}^{l-1}\right) \\
\alpha_{i(t')\to j(t)}^{l} &= \left(\boldsymbol{W}_k \hat{\boldsymbol{h}}_{i(t')}^{l-1}\right)^T \left(\boldsymbol{W}_q \boldsymbol{h}_{j(t)}^{l-1}\right) \cdot \frac{1}{\sqrt{d}}, \quad \hat{\boldsymbol{h}}_{i(t')}^{l-1} = \boldsymbol{h}_{i(t')}^{l-1} + \mathrm{TE}(t'-t) \\
\mathrm{TE}(\Delta t)_{2i} &= \sin\left(\frac{\Delta t}{10000^{2i/d}}\right), \quad \mathrm{TE}(\Delta t)_{2i+1} = \cos\left(\frac{\Delta t}{10000^{2i/d}}\right),
\end{aligned}
\tag{58}
$$

where $\|$ denotes concatenation; $\sigma(\cdot)$ is a non-linear activation function; $d$ is the dimension of node embeddings. The node representation is computed as a weighted summation over its neighbors plus residual connection where the attention score is a transformer-based (Vaswani et al., 2017) dot-product of node representations by the use of value, key, query projection matrices $\boldsymbol{W}_v, \boldsymbol{W}_k, \boldsymbol{W}_q$. Here $\boldsymbol{h}_{j(t)}^{l}$ is the representation of agent $j$ at time $t$ in the $l$-th layer. $i(t')$ is the general index for neighbors connected by temporal edges (where $t' \neq t$) and spatial edges (where $t = t'$ and $i \neq j$). The temporal encoding (Hu et al., 2020) is added to a neighborhood node representation in order to distinguish its message delivered via spatial and temporal edges. Then, we stack $L$ layers to get the final representation for each observation node: $\boldsymbol{h}_i^t = \boldsymbol{h}_{i(t)}^L$. Finally, we employ a self-attention mechanism to generate the sequence representation $\boldsymbol{u}_i$ for each agent as their latent initial states:

$$
\boldsymbol{u}_i = \frac{1}{K}\sum_t \sigma\left(\boldsymbol{a}_i^T \hat{\boldsymbol{h}}_i^t \hat{\boldsymbol{h}}_i^t\right), \quad \boldsymbol{a}_i = \tanh\left(\left(\frac{1}{K}\sum_t \hat{\boldsymbol{h}}_i^t\right)\boldsymbol{W}_a\right),
\tag{59}
$$

where $\boldsymbol{a}_i$ is the average of observation representations with a nonlinear transformation $\boldsymbol{W}_a$ and $\hat{\boldsymbol{h}}_i^t = \boldsymbol{h}_i^t + \mathrm{TE}(t)$. $K$ is the number of observations for each trajectory. Compared with recurrent models such as RNN, LSTM (Sepp and Jürgen, 1997), it offers better parallelization for accelerating training speed and in the meanwhile alleviates the vanishing/exploding gradient problem brought by long sequences. For single-agent Systems, there only left the self-attention mechanism component.

Given the latent initial states, the dynamics of the whole system are determined by the ODE function $g$ which we parametrize as a GNN as in (Huang et al., 2020) for Multi-Agent Systems to capture the continuous interaction among agents. For single-agent systems, we only include self-loop edges in the graph $\mathcal{G} = (\mathcal{V}, \mathcal{E})$, which makes the ODE function $g$ a simple MLP.

We then employ Multilayer Perceptron (MLP) as a decoder to predict the trajectories $\hat{\boldsymbol{y}}_i(t)$ from the latent states $z_i(t)$.

$$\boldsymbol{z}_1(t), \boldsymbol{z}_2(t), \boldsymbol{z}_3(t) \cdots \boldsymbol{z}_N(t) = \text{ODEsolver}(g, [\boldsymbol{z}_1(t_0), \boldsymbol{z}_2(t_0) \cdots \boldsymbol{z}_N(t_0)], (t_0, t_1 \cdots t_K))$$
$$\hat{\boldsymbol{y}}_i(t) = f_{dec}(\boldsymbol{z}_i(t)) \tag{60}$$

## D.2   Implementation Details

**TREAT**

For multi-agent systems, our implementation of TREAT follows GraphODE pipeline. We implement the initial state encoder using a 2-layer GNN with a hidden dimension of 64 across all datasets. We use ReLU for nonlinear activation. For the sequence self-attention module, we set the output dimension to 128. The encoder's output dimension is set to 16, and we add 64 additional dimensions initialized with all zeros to the latent states $z_i(t)$ to stabilize the training processes as in (Huang et al., 2021). The GNN ODE function is implemented with a single-layer GNN from (Kipf et al., 2018) with hidden dimension 128. For single-agent systems, we only include self-loop edges in the graph $\mathcal{G} = (\mathcal{V}, \mathcal{E})$, which makes the ODE function $g$ a simple MLP. To compute trajectories, we use the Runge-Kutta method from torchdiffeq python package s(Chen et al., 2021) as the ODE solver and a one-layer MLP as the decoder.

We implement our model in pytorch. Encoder, generative model, and the decoder parameters are jointly optimized with AdamW optimizer (Loshchilov and Hutter, 2019) using a learning rate of 0.0001 for spring datasets and 0.00001 for *Pendulum*. The batch size for all datasets is set to 512.

TREAT$_{\mathcal{L}_{rev}=\text{gt-rev}}$ and TREAT$_{\mathcal{L}_{rev}=\text{rev2}}$ share the same architecture and hyparameters as TREAT, with different implementations of the loss function. In TREAT$_{\mathcal{L}_{rev}=\text{gt-rev}}$, instead of comparing forward and reverse trajectories, we look at the L2 distance between the ground truth and reverse trajectories when computing the reversal loss.

For TREAT$_{\mathcal{L}_{rev}=\text{rev2}}$, we implement the reversal loss following (Huh et al., 2020) with one difference: we do not apply the reverse operation to the momentum portion of the initial state to the ODE function. This is because the initial hidden state is an output of the encoder that mixes position and momentum information. Note that we also remove the additional dimensions to the latent state that TREAT has. To reproduce our model's results, we provide our code implementation link here.

**LatentODE**

We implement the Latent ODE sequence to sequence model as specified in (Rubanova et al., 2019). We use a 4-layer ODE function in the recognition ODE, and a 2-layer ODE function in the generative ODE. The recognition and generative ODEs use Euler and Dopri5 as solvers (Chen et al., 2021), respectively. The number of units per layer is 1000 in the ODE functions and 50 in GRU update networks. The dimension of the recognition model is set to 100. The model is trained with a learning rate of 0.001 with an exponential decay rate of 0.999 across different experiments. Note that since latentODE is a single-agent model, we compute the trajectory of each object independently when applying it to multi-agent systems.

**HODEN**

To adapt HODEN, which requires full initial states of all objects, to systems with partial observations, we compute each object's initial state via linear spline interpolation if it is missing. Following the setup in (Huh et al., 2020), we have two 2-layer linear networks with Tanh activation in between as ODE functions, in order to model both positions and momenta. Each network has a 1000-unit layer followed by a single-unit layer. The model is trained with a learning rate of 0.00001 using a cosine scheduler.HODEN is a single-agent model, we compute the trajectory of each object independently when applying it to multi-agent systems.

**TRS-ODEN**

Similar to HODEN, we compute each object's initial state via linear spline interpolation if it is missing. As in (Huh et al., 2020), we use a 2-layer linear network with Tanh activation in between as the ODE functions, and the Leapfrog method for solving ODEs. The network has 1000 hidden units and is trained with a learning rate of 0.00001 using a cosine scheduler. TRS-ODEN is a single-agent model, we compute the trajectory of each object independently when applying it to multi-agent systems.

**TRS-ODEN$_{GNN}$**

For TRSODEN$_{GNN}$, we substitute the ODE function in TRS-ODEN with a GraphODE network. The GraphODE generative model is implemented with a single-layer GNN with hidden dimension 128. As in HODEN and TRS-ODEN, we compute each object's missing initial state via linear spline interpolation and the Leapfrog method for solving ODE. For all datasets, we use 0.5 as the coefficient for the reversal loss in (Huh et al., 2020), and 0.0002 as the learning rate under cosine scheduling.

**LGODE**

Our implementation follows (Huang et al., 2020) except we remove the Variational Autoencoder (VAE) from the initial state encoder. Instead of using the output from the encoder GNN as the mean and std of the VAE, we directly use it as the latent initial state. That is, the initial states are deterministic instead of being sampled from a distribution. We use the same architecture as in TREAT and train the model using an AdamW optimizer with a learning rate of 0.0001 across all datasets.

## E  Additional Experiments

### E.1  Comparison of different solvers

We next show our model's sensitivity regarding solvers with different precisions. Specifically, we compare against Euler and Runge-Kutta (RK4) where the latter is a higher-precision solver. We show the comparison against LGODE and TREAT in Table 3.

We can firstly observe that TREAT consistently outperforms LGODE, which is our strongest baseline across different solvers and datasets, indicating the effectiveness of the proposed time-reversal symmetry loss. Secondly, we compute the improvement ratio as $\frac{LGODE-TREAT}{LGODE}$. We can see that the improvement ratios get larger when using RK4 over Euler. This can be understood as our reversal loss is minimizing higher-order Tayler expansion terms (Theoreom 3.1) thus compensating numerical errors brought by ODE solvers.

Table 3: Evaluation results on MSE ($10^{-2}$) over different solvers for multi-agent systems.

| Dataset | *Simple Spring* | | *Forced Spring* | | *Damped Spring* | | *Pendulum* | |
|---|---|---|---|---|---|---|---|---|
| Solvers | Euler | RK4 | Euler | RK4 | Euler | RK4 | Euler | RK4 |
| LGODE | 1.8443 | 1.7429 | 2.0462 | 1.8929 | 1.1686 | 0.9718 | 1.4634 | 1.4156 |
| TREAT | **1.4864** | **1.1178** | **1.6058** | **1.4525** | **0.8070** | **0.5944** | **1.3093** | **1.2527** |
| % Improvement | 19.4057 | 35.8655 | 21.5228 | 23.2659 | 30.9430 | 38.8352 | 10.5303 | 11.5075 |

### E.2  Evaluation across observation ratios.

For LG-ODE and TREAT, the encoder computes the initial states from observed trajectories. To show models' sensitivity towards data sparsity, we randomly mask out 40% and 80% historical observations and compare model performance. As shown in Table 4, when changing the ratios from 80% to 40%, we observe that TREAT has a smaller performance drop compared with LG-ODE, especially on the more complex Pendulum dataset (LG-ODE decreases 22.04% while TREAT decreases 1.62%). This indicates that TREAT is less sensitive toward data sparsity.

Table 4: Results of varying observation ratios on MSE ($10^{-2}$) of multi-agent datasets.

| Dataset | *Simple Spring* | | *Forced Spring* | | *Damped Spring* | | *Pendulum* | |
|---|---|---|---|---|---|---|---|---|
| Observation Ratios | 0.8 | 0.4 | 0.8 | 0.4 | 0.8 | 0.4 | 0.8 | 0.4 |
| LG-ODE | 1.7054 | 1.6889 | 1.7554 | 2.0370 | 0.9305 | 1.0217 | 1.4314 | 1.7469 |
| TREAT | **1.1176** | **1.1429** | **1.3611** | **1.5109** | **0.6920** | **0.6964** | **1.2309** | **1.2110** |

## F  Discussion about Reversible Neural Networks

In literature, there is another line of research about building reversible neural networks (NNs). For example, (Chang et al., 2018) formulates three architectures for reversible neural networks to address

the stability issue and achieve arbitrary deep lengths, motivated by dynamical system modeling. (Liu et al., 2019) employs normalizing flow to create a generative model of graph structures. They all propose novel architectures to construct reversible NN where intermediate states across layer depths do not need to be stored, thus improving memory efficiency.

However, we'd like to clarify that reversible NNs (RevNet) do not resolve the time-reversal symmetry problem that we're studying. The core of RevNet is that input can be recovered from output via a reversible operation (which is another operator), similar as any linear operator $W(\cdot)$ have a reversed projector $W^{-1}(\cdot)$. In the contrary, what we want to study is that the **same operator** can be used for both forward and backward prediction over time, and keep the trajectory the same. That being said, to generate the forward and backward trajectories, we are using the same $g(\cdot)$, instead of $g(\cdot), g^{-1}(\cdot)$ respectively.

In summary, though both reversible NN and time-reversal symmetry share similar insights and intuition, they're talking about different things: reversible NNs make every operator $g(\cdot)$ having a $g^{-1}(\cdot)$, while time-reversible assume the trajectory get from $\hat{z}^{fwd} = g(z)$ and $\hat{z}^{bwd} = -g(z)$ to be closer. Making $g$ to be reversible cannot make the system to be time-reversible.

## G   Impact Statement

This paper presents work whose goal is to advance the field of Machine Learning. TREAT is trained upon physical simulation data (e.g., , spring and pendulum) and implemented by public libraries in PyTorch. During the modeling, we neither introduces any social/ethical bias nor amplify any bias in the data. There are many potential societal consequences of our work, none which we feel must be specifically highlighted here.

