# OpenReview forum: "Physics-Informed Regularization for Domain-Agnostic Dynamical System Modeling"
_NeurIPS.cc/2024/Conference — NeurIPS 2024 poster_

### Official Review · Reviewer_edxC · 2024-07-04

**Soundness:** 3
**Presentation:** 3
**Contribution:** 3
**Rating:** 7
**Confidence:** 3

**Summary:**

The paper introduces a framework named TREAT (Time-Reversal Symmetry ODE) aimed at improving dynamical system modeling through a physics-informed approach. It incorporates Time-Reversal Symmetry (TRS) as a regularization term to enhance model precision. This approach is shown to preserve energy in conservative systems and provide strong inductive bias for non-conservative, reversible systems. Theoretical proofs are provided to demonstrate the numerical benefitd∂. The framework's effectiveness is validated through experiments on nine diverse datasets, showing improvement over existing models.

**Strengths:**

The paper proposed the TRS as a regularization term for dynamical system modeling. this regularization intuitively makes sense, by augmenting the reverse trajectory as additional data for training.

The paper provides theories that TRS minimizes higher-order Taylor terms, enhancing modeling accuracy.

The paper conducted extensive experiments on nine datasets show that TREAT outperforms other neuralODE models.

**Weaknesses:**

1.	The paper only uses graphODE as backbone, from my understanding the TRS regularization can also fit with other neuralODE methods, it would be interesting to add experiments to show its compatibility (e.g. vanilla neuralODE).

2.	The theoretical analysis in theorem 3.1 bounds the error in the embedding space z, however, the loss function in eqn 10 describes the error from original space y. it is worth mentioning the difference due to encoder/decoder error.

**Questions:**

1.	Figure 1 shows the time reserve symmetry system is inclusive in the general classical mechanics. I’m confused regarding the TRS definition in physics and the numerical TRS used in this paper. In my understanding, any dynamical system with deterministic ode description can reverse trace the initial points by solving it from the end point numerically? therefore, the method should be generally applicable?

2.	In figure 4a, do you have any insights why the TREAT does not do well compared with others with shorter prediction length.

---

> ### Author Rebuttal · Authors · 2024-08-07
>
> We sincerely thank you for your valuable comments and suggestions for improving our paper! We would like to make the following claims and hope they can address your concerns.
>
>
> ### **W1. About the use case for Vanilla neural ODE**
>
> Thank you for the valuable suggestion!
> Our TRS loss can be coupled with other models as well, due to its simplicity and flexibility. In our experiments for single-agent systems, TREAT actually becomes neuralODE + TRS loss, as there is no graph structure considered if there’s only one agent (also mentioned in Appendix D.2 page 25 line 673). As shown in Table 1, TREAT consistently outperforms other baselines across diverse single-agent datasets, showing its strong generalization ability and effectiveness.
> Due to its simplicity, we believe other models, such as NeuralODE and flow-based methods, can also benefit from TRS loss.
>
> ### **W2. Theorem 3.1**
> Thanks for pointing this out. We will add an explanation between the original space y and the embedding space z. Our theorem aims to highlight the differences between with and without TRS loss. Both scenarios involve an encoder and a decoder. However, since y and z correspond to each other through the encoder and decoder, the errors introduced by the encoder and decoder do not affect the validity of Theorem 3.1.
>
> ### **Q1. Is TRS a universal property of all classic ODEs?**
> A deterministic ODE can indeed be solved numerically from the endpoint to trace back to the initial points. However, the definition of time-reversal symmetry (TRS) in physics requires that the results of solving the system forward and backward in time must be consistent. This does not always hold for any given ODE function. In Appendix B (page 18, line 523 ~ page 19, line 540), we provide examples of reversible and irreversible systems. These examples illustrate their ODE functions and verify the systems’ energy conservation and time-reversal symmetry. For example, spring systems with frictions do not satisfy TRS, and are also not energy-conservative.
>
> ### **Q2. Explanation of Figure 4 for short-term predictions**
> Thanks for pointing it out! We first observe that on damped springs and forced springs, TREAT offers comparable results, and on the pendulum, TREAT significantly outperforms others.
>
> As for the simple spring with shorter prediction lengths, we assume the potential reasons as follows:
>
> 1. For TRS-ODEN and HODEN, which do not have encoder and decoder structures, we compute each object’s initial state via linear spline interpolation if it is missing. Since Simple Spring is a trivial system, it is less sensitive to the initial state and easier to learn. Also, TRS may be less important for making short-term predictions than long-term predictions as suggested by Figure 4. Therefore, in the short term, forcing TRS and having a more complex encoder structure (TREAT) may add additional burdens to the model learning process, and simple models like TRS-ODEN and HODEN can already work well. However, we can observe in the long run, TREAT significantly achieves better results.
>
> 2. TRS-ODEN and HODEN are two single-agent baselines that do not consider the interactions among agents, i.e., the graph structure. As the dynamics of the Simple Spring are not very complicated, within a short prediction length, the interactions between objects may not have a significant impact on the trajectories. This aligns with observations in existing literature [1].
>
> [1] Zijie Huang, Yizhou Sun and Wei Wang. "Coupled Graph ODE for Learning Interacting System Dynamics." KDD 2021.

---

> > ### Comment · Reviewer_edxC · 2024-08-12
> >
> > I thank authors for the detailed reply for my questions. I agree with most of them.
> >
> > One minor concern regarding the definition of "reversibility" for dynamical system, I think the paper is applicable to all "time reversible" ODEs, which is more general to the hamiltonian reversibility in your appendix B? as long as the function is deterministic say $\dot{x}=f(x)$, it can always be back tracked by  $\dot{x}=-f(x)$? this is also applicable to the spring system with friction.

---

> ### Author Response · Authors · 2024-08-12
>
> Thank you very much for your valuable feedback!
>
> Regarding your minor concern, we made the following clarification:
>
> The definition of TRS is $\frac{d R\circ x}{dt} = -F(R\circ x)$ as shown in page 4 line 111-120. For Hamiltonian systems, the state x contains both position q and velocity p, i.e. $x = (q,p)$, and the reversing operator would work as $R\circ (q,p) = (q,-p)$ according to [1]. This is can simply be understood as: if a pendulum swings from A to B (with no frictions and no external forces, etc), we can just revert its velocity in the opposite direction, and let it swing back from B to A, the two trajectories would be the same. In this case, only velocity is reversed and position at B is kept the same.
>
> Now let's check for spring systems with frictions, would it satisfy $\frac{d R\circ x}{dt} = -F(R\circ x)$. As shown in Appendix B.3, the deterministic ODE functions for such systems are defined as $\frac{dq}{dt} = \frac{p}{m}$, $\frac{dp}{dt} = -kq-\gamma\frac{p}{m}$, where $m$ is the mass. Since $R\circ (q,p) = (q,-p)$, going back to the TRS definition $\frac{d R\circ x}{dt} = -F(R\circ x)$, we need to check for position $q$, does the first ODE function follow $\frac{dq(-t)}{d(-t)} = -F(q(-t), p(-t))$, which implies $\frac{dq(t)}{dt} = -F(q(t), -p(t))$. Similarly, for velocity $p$, we need to check does the second ODE function follows $\frac{dp(-t)}{d(-t)} = -F(q(-t), p(-t))$, which implies $\frac{dp(t)}{dt} = F(q(t), -p(t))$.
>
> We can easily verify that the first ODE function holds ($\frac{dq(t)}{dt} = -F(q(t), -p(t))$), while the second ODE function fails ($\frac{dp(t)}{dt} = F(q(t), -p(t))$). Therefore, spring systems with frictions do not satisfy TRS.
>
> Please kindly let us know if there are any questions further, for which we're happy to provide additional information and clarifications. We truly appreciate the time and effort you’ve invested in reviewing our work!
>
> [1] Lamb, J. S., & Roberts, J. A. (1998). Time-reversal symmetry in dynamical systems: a survey. Physica D: Nonlinear Phenomena, 112(1-2), 1-39.

---

> > ### Comment · Reviewer_edxC · 2024-08-12
> >
> > thanks for the clarification, I mistook "time reversibility" vs "time-reversal symmetry" in your paper. i will raise my score to 7.

---

> > > ### Author Response · Authors · 2024-08-12
> > >
> > > Thank you for your prompt response and positive feedback on our rebuttal! We will definitely incorporate your valuable suggestions in our revised version.

---

### Official Review · Reviewer_D3dA · 2024-07-08

**Soundness:** 3
**Presentation:** 2
**Contribution:** 3
**Rating:** 6
**Confidence:** 4

**Summary:**

The paper proposes a novel Time-Reversal Symmetry (TRS) graph neural ODE, where the TRS is introduced as a soft regularisation term.

**Strengths:**

The paper presents a novel method to combat numerical errors for graph neural ODEs.

**Weaknesses:**

The paper does not introduce early enough that the time-reversal symmetry (TRS) method has been applied to other DL methods such as the TRS-ODEN.

**Questions:**

Are there cases where the time-reversal symmetry expectation is violated?

A comparison to a numerical solution could serve as an additional benchmark to show how TRS decreases the numerical errors and improves the prediction. Also, why considering only MSE as a performance metric?

The code is anonymised, but it really needs cleaning and proper documentation to the very least docstrings and comments; I have not tested running the code.

The work will benefit from additional proofreading.

**Limitations:**

The authors do not provide a separate Limitations section, and do not discuss limitations of the work in the Conclusions section.

---

> ### Author Rebuttal · Authors · 2024-08-07
>
> We're grateful for your support and helpful feedback! Please kindly find our response below to the questions raised in the review and let us know if you have any further questions.
>
> ### **W1: Reference about TRS-ODEN.**
> Thank you for your advice! We would like to discuss more about TRS-ODEN in the introduction part in the revised version. However, we would like to clarify a few points:
> We do not position our paper as an improved version of how to enforce TRS as in TRS-ODEN paper. Instead, we would like to find a general physics-informed prior for domain-agnostic dynamical system modeling. To this end, we found that TRS is a great property to achieve that. We would like to emphasize our major contribution is to find the numerical benefits of TRS regularization, i.e. minimizing higher-order Taylor expansion terms during ODE integration steps. This makes the proposed TRS loss widely applicable to a wide range of dynamical systems, regardless of their physical properties (i.e. even irreversible systems). Therefore, we did not see TRS-ODEN as a direct competitor.  Nonetheless, in sections where details are needed, we discussed more details about TRS-ODEN (first appeared in Sec.2).  However, we will make sure to clarify this further in the introduction part to make it clearer.
>
> ### **Q1: Cases where TRS is violated**
> Thanks for your question! Indeed, some systems may not strictly obey the TRS due to situations such as time-varying external forces, internal friction, and underlying stochastic dynamics. For example, spring systems with frictions do not adhere to TRS. We illustrate examples for reversible and irreversible systems in Appendix B (page 18, line 523 ~ page 19, line 540).  Therefore, a desired model shall be able to flexibly inject time-reversal symmetry as a soft constraint, so as to cover a wider range of real-world dynamical systems. Note that from the numerical aspect, we also theoretically prove that the TRS loss effectively minimizes higher-order Taylor expansion terms during ODE integration, offering a general numerical advantage for improving modeling accuracy across a wide array of systems, regardless of their physical properties (even for irreversible systems). Therefore, TREAT achieves high-precision modeling from both aspects as depicted in Figure 1(a).
>
> ### **Q2-1: Can TRS decrease numerical errors across different solvers?**
> To show TRS can in general help to reduce numerical errors across different choices of solvers, we show the performance of the Euler method and Runge-Kutta (RK4) methods, which have different trade-off between precision and time. As suggested by the results below (also shown in Appendix E.1 on page 26, line 726-735), TREAT consistently outperforms LGODE, our strongest baseline, across different solvers and datasets.
> We also notice that the improvement ratio $\(\frac{LGODE - TREAT}{LGODE}\)$ is larger when using RK4 compared to Euler. This suggests that our TRS  loss minimizes higher-order Taylor expansion terms, thus compensating for numerical errors introduced by ODE solvers.
>
> | Dataset &nbsp;&nbsp; &nbsp;&nbsp;  &nbsp; &nbsp; &nbsp; &nbsp; | Simple Spring   &nbsp;  &nbsp;&nbsp;&nbsp;| Forced Spring &nbsp;&nbsp;&nbsp;| Damped Spring  &nbsp;&nbsp;&nbsp;     | Pendulum                  |
> |---------------|-----------------------|---------------|---------------------|--------------------|
>
> Solvers| &nbsp; Euler   |  &nbsp;  RK4   | &nbsp; Euler   |  &nbsp;  RK4   | &nbsp; Euler  |  &nbsp;  RK4  | &nbsp; Euler   |  &nbsp;  RK4
> :--|:--|:--|:--|:----|:---|:-----|:---------|:-----------
> LGODE         | 1.8443  | 1.7429      | 2.0462  | 1.8929      | 1.1686  | 0.9718      | 1.4634  | 1.4156
> TREAT         | **1.4864** | **1.1178** | **1.6058** | **1.4525** | **0.8070** | **0.5944** | **1.3093** | **1.2527**
> \% Improvement | 19.4057 | 35.8655     | 21.5228 | 23.2659     | 30.9430 | 38.8352     | 10.5303 | 11.5075
>
> ### **Q2-2: Evaluation metric**
> Thank you for your valuable feedback. Since our work focuses on deterministic trajectory prediction, calculating the MSE between the predicted trajectory and the ground truth trajectory is the most direct and effective measure. Therefore, we followed the experimental settings of existing work [1] [2].
>
> [1]Huang, Z., et al.  "Learning continuous system dynamics from irregularly-sampled partial observations." Advances in Neural Information Processing Systems 2020
>
> [2] Kipf, Thomas, et al. "Neural relational inference for interacting systems." International conference on machine learning. PMLR, 2018.
>
> ### **Q3: Code**
> Thank you for your feedback. We apologize for not providing sufficient docs and comments for the code. We have refined our code structure and documentation in the anonymous GitHub repo.
>
> ### **Q4: Discussion about limitations**
> Thank you for your valuable suggestion. We have included the limitations section in Appendix H (page 27, lines 776-779). We apologize for any inconvenience this may have caused. In the revised version, we promise to include it in the main part of the paper.

---

> > ### Comment · Reviewer_D3dA · 2024-08-12
> >
> > I would like to thank the authors for looking into the provided comments and suggestions, and I looking forward to reading the revised manuscript.

---

> > > ### Author Response · Authors · 2024-08-12
> > >
> > > Dear Reviewer D3dA,
> > >
> > > We sincerely thank you for your encouraging feedback on our rebuttal! We hope this align with your expectations and positively influence the score. We will definitely incorporate your valuable suggestions in our revised version, and please kindly let us know if there's anything else for further improvement. Thanks again for your valuable time and efforts in reviewing our paper!

---

### Official Review · Reviewer_SfDm · 2024-07-11

**Soundness:** 3
**Presentation:** 3
**Contribution:** 3
**Rating:** 6
**Confidence:** 3

**Summary:**

This paper proposes a regularization term to enforce Time-Reversal Symmetry (TRS) for modeling dynamical systems. The method helps preserve energies for conservative systems while serving as a strong inductive bias for non-conservative reversible systems. They also prove that TRS loss can universally improve modeling accuracy by minimizing higher-order Taylor terms in ODE integration. Numerical examples show that integrating the TRS loss within neural ordinary differential equation models demonstrates superior performance on diverse physical systems.

**Strengths:**

This paper focuses on learning complex physical dynamics from data and incorporating physics priors into learning models, which is an important area. This paper proposes a regularization term to enforce Time-Reversal Symmetry (TRS), which is an important physics prior.
Also, the paper numerically shows that integrating the TRS loss within neural ordinary differential equation models demonstrates superior performance on diverse physical systems.

**Weaknesses:**

While the paper on time-reversal symmetry demonstrates several strengths, there are also some potential weaknesses that could be addressed:

1.  Time-Reversal Symmetric ODE Network has been proposed (https://arxiv.org/pdf/2007.11362). This article is also cited in this paper. However, this paper only points out that two articles use different properties (Equation 5 and Lemma 2.1) to sign the regularization term, the difference in the final regularization term is not fully explained.

2. The numerical experiments also demonstrated that the method proposed in this paper outperforms existing time-reversal symmetry regularization (https://arxiv.org/pdf/2007.11362). However, the reason why this method is better in this paper is not explained in detail.

3. Some descriptions in the text are not clear enough, please refer to the question section below.

**Questions:**

1. In the proof of Lemma 2.1, it is required that R be an involution operator. However, such assumption is not stated in Lemma 2.1. Is there any evidence to prove that this assumption is reasonable？

2. The definition of R does not appear in the paper. Can the method proposed in this paper hold for any R?

3. How is regularization term 9 obtained based on Lemma 2, as R did not appear in the final regularization term 9？

4. Does Theorem 3.1 hold for any untrained NN?

5. The definition of $L_{reverse2}$ does not appear in the paper.  Can the condition $L_{reverse2}=L_{reverse}$ be satisfied?

**Limitations:**

Yes

---

> ### Author Rebuttal · Authors · 2024-08-07
>
> We thank you for your insightful comments on improving our paper. However, we believe there’s some misunderstanding and would like to make the following claims to address your concerns.
>
> ### **W1: The difference to the final implementations.**
> We would like to mention that both methods are approximations towards TRS. In Appendix A.4 (page16), we show the difference of the final implementations and their visualizations in Figure 8 (assuming one integration step).
>
> The TRS loss in TREAT (ours) follows Lemma 2.1: $R \circ \Phi_t \circ R \circ \Phi_t = \text{I} $
> It means: we start from $\boldsymbol{\hat{y}}_i ^{\text{fwd}}(0)$ and first move forward one step, reaching the state $\boldsymbol{\hat{y}}_i ^{\text{fwd}}(1)$. Then we reverse it and move forward one step in the opposite direction, getting $\boldsymbol{\hat{y}}_i ^{\text{rev}}(0)$. Finally, we reverse it and it shall restore back to the same state. That is ideally it should be the same as $\boldsymbol{\hat{y}}_i ^{\text{fwd}}(0)$.
>
> The second reverse loss in TRS-ODEN follows Eqn 5 as $R \circ \Phi_t = \Phi_{-t}\circ R $. It means we first reverse the initial state and move forward one step in the opposite direction to reach $ \boldsymbol{\hat{y}}_i ^{\text{rev2}}(-1)$. We then perform a symmetric operation to reach $ \boldsymbol{\hat{y}}_i ^{\text{rev2}}(1) $, which should align with the forward one $\boldsymbol{\hat{y}}_i ^{\text{fwd}}(1)$.
>
> The key differences are illustrated in Figure 8. Our method forces the starting point of backward trajectories to be the same as the end point of the forward one. In TRS-ODEN, the backward trajectories start from the same point as the forward one.
> We analytically show why our implementation is better based on Figure 8 in Appendix A.4.
>
> ### **W2: Why our implementation is better.**
> As shown in Appendix A.4 (page 16 line 494-501), we analytically show our implementation based on Lemma 2.1 to approximate TRS has a lower maximum error compared to TRS-ODEN, supported by empirical experiments in Sec. 4.2.
>
> We here use one integration step for illustration purpose. Specifically, if we assume the two reconstruction losses
> are of the same value $a$, and the two TRS losses have reached the same value $b$, we show that the maximum error between the reversal and ground truth trajectory for each agent, made by TREAT is smaller.  In TREAT, the maximum error is $max(a,b)$, whereas TRS-ODEN is $a+b$.
>
> Finally, we would like to summarize our contribution again: we do not position our paper as an improved version of how to enforce TRS as in the TRS-ODEN paper. Instead, we would like to find a general physics-informed prior for domain-agnostic dynamical system modeling. To this end, we found that TRS is a great property to achieve that. Our major contribution is a physics-inspired regularizer that works well beyond physics domains, due to its numerical properties. Specifically, our TRS loss minimizes higher-order Taylor expansion terms during ODE integration steps. This makes the proposed TRS loss widely applicable to a wide range of dynamical systems, regardless of their physical properties.
>
> ### **Q1.**
> Thanks for your question. The reversing operator R is an involution by definition [1], i.e. $R\circ R = I$. We will clarify this in Lemma 2.1 based on your valuable suggestion.
>
> For example:
> Consider reversing the position $q(t)$, we have $R\circ q(t) = q(t)$ [1]. Therefore we can get $R\circ R\circ q(t)=R \circ q(t) = q(t) $,
>
> As for velocity $p(t)$,  we have $R\circ p(t) = -p(t)$ [1]. Therefore we can also get $R\circ R\circ p(t)=R\circ (-p(t))= p(t)$
>
> [1] Lamb, J. S., & Roberts, J. A. (1998). Time-reversal symmetry in dynamical systems: a survey. Physica D: Nonlinear Phenomena, 112(1-2), 1-39.
>
> ### **Q2.**
> The definition of $R$ is illustrated in section 2.2 (page 4 line 111-128). It is a reversing operator defined as $R: {x} \mapsto R \circ {x}, R \circ {x} (t) = {x}(-t)$ where $x$ is the observational state of the system. For different systems, $x$ can contain different state variables, so this definition is universal to all systems. For example, if $x = (q,p)$ where $q$ is the location and $p$ is the velocity, $R\circ x = (q,-p)$ [1].
>
> [1] Lamb, J. S., & Roberts, J. A. (1998). Time-reversal symmetry in dynamical systems: a survey. Physica D: Nonlinear Phenomena, 112(1-2), 1-39.
>
> ### **Q3.**
> As shown in Figure 2 and Figure 8, $z^{\text{rev}}(t’_0)=R \circ z^{\text{fwd}}(t_K)=z^{\text{fwd}}(t_K)$.
> Combining this with Eqn 8, we obtain $ y^{\text{rev}}(t’_j)$
> and finally, we derive Eqn 9.
>
> ### **Q4.**
> Yes. Theorem 3.1 does not restrict the form of the ODE function $g$. The ODE function $g$ can be any NN in practice.
>
> ### **Q5.**
> Thanks for pointing this out! The definition of $\mathcal{L}_{reverse2}$ is proposed in TRS-ODEN, and detailed in our Appendix A.4. We will take your advice and add it to the main text for clarification.

---

> > ### Comment · Reviewer_SfDm · 2024-08-12
> >
> > Thanks for the response, my questions are basically resolved and the score has been updated. The main concern is that novelty migt be slightly limited, because time-reversible symmetry and adding it  to the network through regularization have been proposed. I fully understand the difference in the details of the two methods after reading the response, and I feel a possible improvement is to theoretically prove that the existing methods cannot meet the good properties of the NN in the paper.

---

> > > ### Author Response · Authors · 2024-08-12
> > >
> > > Dear Reviewer SfDm,
> > >
> > > We sincerely appreciate your valuable feedback and raising the score! We will definitely incorporate your suggestions into our revised version. Regarding the novelty, we would like to further clarify that: our work is not a simple enhancement of achieving TRS. Our motivation is to find a domain-agonistic physical prior and achieve high-precision modeling for a wide range of dynamical systems, in contrast to the domain-specific physical prior such as energy conservation. Our key contribution is that: **While TRS is a domain-specific physical prior**, we present **the first theoretical proof** that **TRS loss can universally improve modeling accuracy** by minimizing higher-order Taylor terms in ODE integration, which is numerically beneficial to various systems regardless of their properties, even for irreversible systems. **This bridges the specific physical implication and the general numerical benefits of the physical prior - TRS (as illustrated by Figure 1 (a).**
> > >
> > > Regarding the model performance, our Lemma 3.2 demonstrates that when both methods (TREAT and TRS-ODEN) achieve the same numerical errors $\( L_{\text{pred}} \)\quad and \quad \( L_{\text{reverse}} \)$, the maximum error between the reversal and ground truth trajectory for each agent made by our model, TREAT, is smaller compared to the TRS-ODEN. This is also validated by our experiments in Table 1 ( ablation study of $TREAT\_{\text{Lrev=rev2}}$.
> > >
> > > Once again, we sincerely appreciate your time and positive feedback for improvement! Please kindly let us know if you have any questions further.

---

> > > > ### Author Response · Authors · 2024-08-13
> > > >
> > > > Dear Reviewer SfDm,
> > > >
> > > > We really appreciate your time and efforts in reviewing our submission again! With the author-reviewer discussion period drawing close to the end, we sincerely hope to ensure that we have adequately addressed your newest questions regarding the novelty. We would like to inquire if there are any concerns about our rebuttal, for which we're happy to provide additional information and further clarifications.
> > > >
> > > > Once again, we truly appreciate the time and effort you’ve invested in reviewing our work!

---

> > > > > ### Comment · Reviewer_SfDm · 2024-08-14
> > > > >
> > > > > Thank you for your response. My concerns and questions have been adequately addressed. I keep my positive score.

---

> > > > > > ### Author Response · Authors · 2024-08-14
> > > > > >
> > > > > > Dear Reviewer SfDm,
> > > > > >
> > > > > > Thank you so much for your positive feedback! We sincerely appreciate your valuable time and efforts in reviewing our submission.

---

### Official Review · Reviewer_hqsc · 2024-07-13

**Soundness:** 3
**Presentation:** 4
**Contribution:** 4
**Rating:** 8
**Confidence:** 4

**Summary:**

This paper proposes a method to enhance neural ordinary differential equations (ODEs) by enforcing approximate Time-Reversal Symmetry. A self-supervised regularization term is introduced to align forward and backward trajectories predicted by a neural network, promoting energy conservation and stability in the system. Experiments are conducted on 9 datasets, comprising both real-world and simulated systems. The proposed approach outperforms other baseline methods, demonstrating its effectiveness in improving neural ODEs.

**Strengths:**

* The idea of imposing time-reversal symmetry via a regularization term is novel and well-motivated.
* The implementation of the regularization term is simple yet the effect is visible.
* Overall the article is well written, and technical details are explained clearly.
* The results show significant improvement over other baseline models.

**Weaknesses:**

* It would be more interesting to see if the model works for other real-world examples/experiments.
* The article lacks a more comprehensive review on other  energy-preserving / time-reversal neural ODE solvers.

**Questions:**

Does the method work well with models other than GraphODE?

**Limitations:**

The authors briefly touched upon the possibility of incorporating properties in the spatial aspect such as translation and rotation equivariance.

---

> ### Author Rebuttal · Authors · 2024-08-07
>
> We sincerely appreciate your insightful comments and valuable advice on improving our paper. We highly appreciate your recognition of the novelty and significance of our work. Regarding your questions, we detailed our responses below.
>
> ### **W1: Empirical results on additional real-world examples.**
> We appreciate your suggestions to add more real-world examples. In our experiments, we have 9 diverse datasets spanning across 1.) single-agent, multi-agent systems; 2.) simulated and real-world systems; and 3.) systems with different physical priors. Most of them are simulated datasets, as real-world datasets do not conform neatly to specific physical laws, making it challenging to classify them as either conservative or reversible straightforwardly and to compare them with existing physics-informed baselines. Notably, on one real-world human motion dataset (walking object), TREAT outperforms other baselines significantly, showing case its strong generalization ability and numerical benefits.
>
>
> To address your concern, we additionally added a new human motion dataset (dancing object). The results below suggest that TREAT consistently outperforms other baselines on the new real-world dataset, showing its effectiveness.
>
> |model | TREAT | LGODE | TRS_ODEN |HODEN|LatentODE|
> |--|--|--|--|--|--|
> |MSE|$\underline{2.5420}$|2.7270|3.6885|4.4342|23.0157|
>
> ### **W2: review on other energy-preserving/time-reversal neural ODE solvers.**
>
> We appreciate your suggestion to incorporate more related work on energy-preserving and time-reversal neural ODE solvers. In response, we have drafted the following additions to our revised manuscript.
>
> Various methods have been developed to maintain the total energy of dynamic systems over time [1][2][3][4]. However, strictly energy-conservative approaches can be unrealistic for non-isolated systems that interact with their environments. Conversely, methods that allow for both energy-conserving and dissipative models, as well as reversible and irreversible systems, offer more flexibility [5][6][7][8]. These methods, however, require prior knowledge of the system's properties. For example, [7] necessitates determining the appropriate bracket operator for different systems. Our model, in contrast, introduces a unified approach that learns both the trajectories and the aforementioned properties dynamically, without needing to assign a specific property beforehand.
>
>
> [1] Greydanus, Samuel, Misko Dzamba, and Jason Yosinski. "Hamiltonian neural networks." Advances in neural information processing systems 32 (2019).
>
> [2] Gruver et al, Deconstructing the inductive biases of Hamiltonian neural networks, ICLR 2022.
>
> [3]Han, Chen-Di, et al. "Adaptable Hamiltonian neural networks." Physical Review Research 3.2 (2021): 023156.
>
> [4] Mattheakis, Marios, et al. "Hamiltonian neural networks for solving equations of motion." Physical Review E 105.6 (2022): 065305.
>
> [5] Zhong et al, Dissipative symoden: Encoding Hamiltonian dynamics with dissipation and control into deep learning, et al, ICLR 2020 workshop.
>
> [6] Morrison, Philip J. "A paradigm for joined Hamiltonian and dissipative systems." Physica D: Nonlinear Phenomena 18.1-3 (1986): 410-419.
>
> [7] Gruber, et al, Reversible and irreversible bracket-based dynamics for deep graph neural networks, NeurIPS 2023.
>
> [8] Huh et al, Time-reversal symmetric ode network, NeurIPS 2020.
>
> ### **Q1. Can the TRS loss be coupled with other models?**
> Our TRS loss can be coupled with other models as well, due to its simplicity and flexibility. In our experiments for single-agent systems, TREAT actually becomes neuralODE + TRS loss, as there is no graph structure considered if there’s only one agent. As shown in Table 1, TREAT consistently outperforms other baselines across diverse single-agent datasets, showing its strong generalization ability and effectiveness.
> Due to its simplicity, we believe other models, such as NeuralODE and flow-based methods, can also benefit from TRS loss.

---

### Author Response · Authors · 2024-08-07
**General Response to all Reviewers**

We sincerely thank all reviewers for their valuable time and insightful feedbacks! We acknowledge the positive comments such as "Novel approach and well-motivated " (Reviewer hqsc, D3dA), “Simple-yet-effective approach and superior performance” (Reviewer hqsc, SfDm,edxC), "Well-written" (Reviewer hqsc), "Theoretical Analysis” (Reviewer edxC). We have also responded to all the questions point by point and will incorporate these suggestions into our revision to enhance our paper.

---

### Decision · Program_Chairs · 2024-09-25

**Decision:**

Accept (poster)

**Comment:**

This paper introduces Time-Reversal Symmetry (TRS) as a regularization term to improve NODEs. The TRS regularization is designed to align forward and backward trajectories predicted by a NODE, promoting energy conservation and stability in the system. The paper is well-motivated, and the proposed method is both innovative and relevant, with theoretical justifications provided to support the benefits of TRS in improving modeling accuracy. The approach is validated with a number of experiments across nine diverse datasets, demonstrating good performance compared to baseline models. In summary, the paper is fun to read and stimulating.

Given the strengths of the paper (i.e., the novel contribution, nice theoretical foundations, and experimental validation) and the very positive reviews, I recommend accepting this paper. However, I urge the authors to carefully revise the manuscript when preparing the camera ready version. Specifically, I suggest to expand the empirical scope and providing clearer theoretical explanations, as well as to address all the discussed issues during the rebuttal phase.